# Pitavastatin activates mitophagy to protect EPC proliferation through a calcium-dependent CAMK1-PINK1 pathway in atherosclerotic mice

Jie Yang[1,2,3], Mengjia Sun[1,2,3], Ran Cheng[1,2,3], Hu Tan[1,2], Chuan Liu[1,2], Renzheng Chen[1,2], Jihang Zhang[1,2], Yuanqi Yang[1,2], Xubin Gao[1,2] & Lan Huang [1,2✉]

Statins play a major role in reducing circulating cholesterol levels and are widely used to prevent coronary artery disease. Although they are recently confirmed to up-regulate mitophagy, little is known about the molecular mechanisms and its effect on endothelial progenitor cell (EPC). Here, we explore the role and mechanism underlying statin (pitavastatin, PTV)-activated mitophagy in EPC proliferation. $ApoE^{-/-}$ mice are fed a high-fat diet for 8 weeks to induce atherosclerosis. In these mice, EPC proliferation decreases and is accompanied by mitochondrial dysfunction and mitophagy impairment via the PINK1-PARK2 pathway. PTV reverses mitophagy and reduction in proliferation. *Pink1* knockout or silencing *Atg7* blocks PTV-induced proliferation improvement, suggesting that mitophagy contributes to the EPC proliferation increase. PTV elicits mitochondrial calcium release into the cytoplasm and further phosphorylates CAMK1. Phosphorylated CAMK1 contributes to PINK1 phosphorylation as well as mitophagy and mitochondrial function recover in EPCs. Together, our findings describe a molecular mechanism of mitophagy activation, where mitochondrial calcium release promotes CAMK1 phosphorylation of threonine[177] before phosphorylation of PINK1 at serine[228], which recruits PARK2 and phosphorylates its serine[65] to activate mitophagy. Our results further account for the pleiotropic effects of statins on the cardiovascular system and provide a promising and potential therapeutic target for atherosclerosis.

[1] Institute of Cardiovascular Diseases of PLA, the Second Affiliated Hospital, Army Medical University (Third Military Medical University), Chongqing, China. [2] Department of Cardiology, the Second Affiliated Hospital, Army Medical University (Third Military Medical University), Chongqing, China. [3] These authors contributed equally: Jie Yang, Mengjia Sun, Ran Cheng. ✉email: huanglan260@126.com

Vascular endothelial injury contributes to the development of major cardiovascular diseases (CVDs), and promoting re-endothelialization after arterial injury is critical for their prevention. Circulating endothelial progenitor cells (EPCs) mobilize from the bone marrow and home to the sites of damaged vascular tissue to re-establish an intact endothelial layer[1–3]. The level of circulating EPCs and measures of their function are used diagnostically to predict the prognosis of CVDs[4–6]. Enhancing the proliferation and other functions of EPCs could, therefore, promote the recovery of endothelial integrity and ameliorate prognosis in CVDs.

Pharmacological agents that stimulate EPCs have aroused great interest. Among them, statins are considered the most effective and safe[7–9]. Statins reduce cholesterol biosynthesis by inhibiting 3-hydroxyl-3-methyl coenzyme A (HMG CoA) reductase, are widely used to treat hyperlipidemia, and are primarily used for CVD prevention. Besides inhibiting cholesterol, statins decrease the production of non-steroidal isoprenoid compounds. Beyond lipid reduction, statins elicit pleiotropic effects across cell types, including anti-inflammatory, antioxidant, anti-apoptotic, and antithrombotic effects[10–12]. Recently, atorvastatin was found to protect mesenchymal stem cells (MSCs) from hypoxia and serum deprivation by activating autophagy[13]; however, it can cause side effects such as increased blood glucose levels. Statin-induced autophagy plays a critical role in diabetogenesis through hepatic gluconeogenesis[14]. Our recent study revealed that store-operated calcium entry (SOCE)-induced autophagy protects EPC proliferation during ox-LDL exposure[15], providing potential evidence that statin-activated autophagy or mitophagy is related to EPC regulation. The mechanism of action of statins in autophagy or mitophagy induction remains unclear.

Atherosclerosis is a chronic progressive disease with the pathological changes in oxidative stress, immune response, and lipid metabolism, such as uncontrolled accumulation of lipids caused by desialylated lipoproteins[16]. Increasing evidences reveals that mitochondrial dysfunction also contributes to atherosclerosis. In the early stage of atherosclerosis, increased production of reactive oxygen species (ROS) in mitochondria, accumulation of mitochondrial DNA (mtDNA) damage, and progressive respiratory chain dysfunction, resulted in endothelial cells (ECs) dysfunction and vascular smooth muscle cells (VSMCs) phenotypic conversion[17]. After a long term, ECs apoptosis, VSMCs phenotypic conversion, and inflammatory cells infiltration further promoted the development of atherosclerosis and led to vulnerable plaque in the end. Mitophagy is the only mechanism to eliminate these mutant or damaged mitochondria and maintains mitochondrial homeostasis and energy metabolism. Recently, multiple mitophagy programs that operate independently or undergo crosstalk have been revealed, which function through modulated autophagy receptor activities at the mitochondrial outer membrane (OMM). OMM-localized receptors are ubiquitylated by the E3 ligase PARK2 to recruit ubiquitin-binding autophagy receptors such as NBR1, OPTN, and SQSTM1, which then can attach to autophagosomes via their LC3 interacting region (LIR) motif[18–20]. OMM-anchor proteins that contain LIR motifs such as BNIP3L/Nix and FUNDC1, regulate mitophagy by phosphorylation[21]. Mitochondrial membrane potential (MMP) dysfunction is observed in statin-induced myopathy, which contributes to pathogenesis[22,23]. Statins might therefore activate mitophagy through OMM protein ubiquitylation, phosphorylation, and/or binding with autophagosomes, accounting for their pleiotropic effects.

Intracellular calcium concentration ($[Ca^{2+}]_i$) change as an autophagic regulator is a controversial subject. Most evidence indicates that elevated $[Ca^{2+}]_i$ levels increase autophagy[15,24,25], but other reports suggest that inositol 1,4,5-trisphosphate receptor (IP3R)-driven calcium signaling suppresses autophagy[26,27]. Mitochondria are an important intracellular calcium store and actively participate in calcium signaling. Interestingly, statins influence skeletal muscle fibers and B lymphocyte calcium homeostasis through mitochondrial calcium extrusion[28,29]. However, the action of mitochondrial calcium efflux channels is still under debate, as our laboratory has illustrated that calcium signaling regulates EPC autophagy[15]. We proposed that statins might elicit mitochondrial calcium release in EPCs and be associated with mitophagy induction. Here, we investigated EPC mitophagy and proliferation in atherosclerotic mice, as well as the role of pitavastatin (PTV) in mitophagy induction and EPC proliferation.

## Results

### EPC proliferation inhibition and mitochondrial dysfunction in atherosclerotic mice

To establish atherosclerotic mice, we fed $ApoE^{-/-}$ mice with a high-fat diet (HFD, 21% fat, and 0.15% cholesterol) for 8 or 16 weeks (Supplementary Fig. 1a). We isolated aortas from mice fed HFD for 8 weeks (HFD8w), where atherosclerotic lesions appeared clearly compared to those on a normal diet (ND). More lesions were observed after 16 weeks of HFD (HFD16w) (Supplementary Fig. 1b). Immunostaining of the lesions showed CD68 positive cells, thus represented macrophage-rich in HFD8w mice atherosclerosis lesions and vascular wall (Supplementary Fig. 2). Additionally, we focused on the immunity state of the established atherosclerotic mice. Atherosclerosis markedly increased levels of myeloid cell subsets with the highest proportion of neutrophils (Supplementary Fig. 1c). The elevated levels of circulating neutrophils pointed to enhanced myeloid cell supplied by the bone marrow. Likewise, we observed higher levels of the upstream progenitor cells. Compared to control, atherosclerotic mice marrow had significantly more common myeloid progenitor cells (CMP, c-Kit+ CD41+), granulocyte macrophage progenitor cells (GMP, c-Kit+ CD64 + CD16/32+), common lymphoid progenitor cells (CLP, c-Kit+ CD127 + CD93+) and hematopoietic stem and progenitor cells (HSPC, c-Kit+ Sca+ lineage-) (Supplementary Fig. 1d).

EPC proliferation from mice on different diets was evaluated by a new approach of real-time cell analyzer (RTCA) and a traditional method of cell counting kit-8 (CCK-8). RTCA results showed that the normalized cell index of EPCs significantly decreased in HFD8w and HFD16w mice (Fig. 1a). CCK-8 assays revealed that the absorbance intensities decreased by 25.24% in HFD8w mice compared to that in ND mice, and this effect was pronounced in HFD16w mice with a 47.31% reduction (Fig. 1b). MMP in HFD EPCs decreased significantly (Fig. 1c), as indicated by a decrease in the intensity ratio of red to green JC-1 fluorescence compared with that in ND EPCs (Fig. 1e). To further explore the mitochondrial function of EPCs in atherosclerosis, we applied a mitochondrial-targeted fluorescent superoxide sensor. HFD EPCs displayed increased mitochondrial superoxide generation, measured by MitoSOX red fluorescence, indicating ROS accumulation in atherosclerotic mice EPCs (Fig. 1d, f). Moreover, we observed swollen mitochondria with distorted cristae in HFD EPCs than in ND EPCs using TEM (Fig. 1g). Together, this confirmed that EPC proliferation decreased with mitochondrial dysfunction in atherosclerotic mice.,

### Mitophagy is impaired in EPCs from atherosclerotic mice

Mitophagy serves as a safeguard in maintaining mitochondrial homeostasis and dynamic. Mitophagic defect or inhibition has been confirmed to accelerate disease progression and worsen the outcome in multiple animal models of CVD[15]. Mitophagy impairing is usually accompanied by mitochondrial

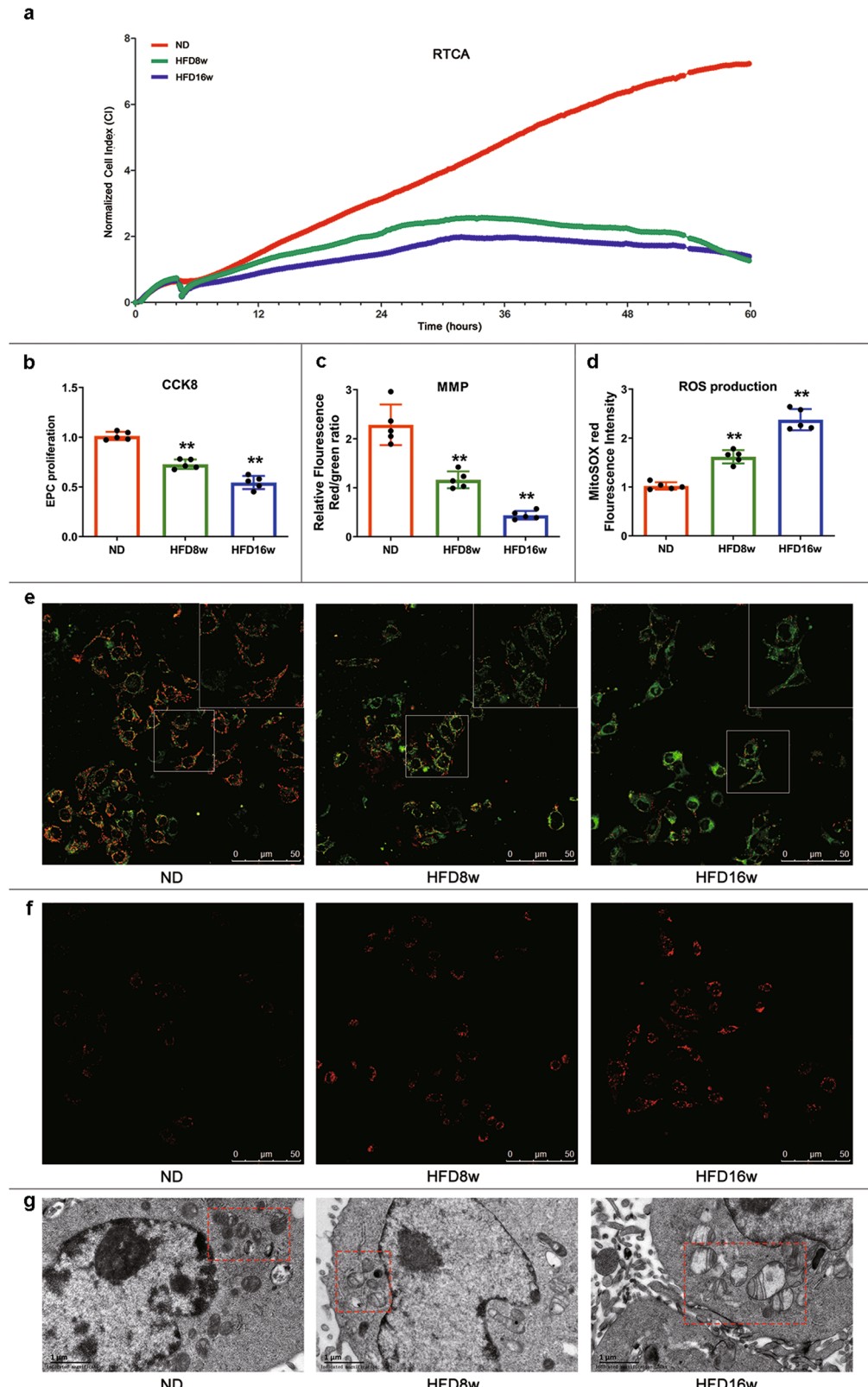

accumulation, dysfunction, and morphology disorder. Therefore, we examined the mitophagic markers of EPCs from athero-sclerotic mice. As shown in Fig. 2a–c, HFD8w significantly decreased the turnover of MAP1LC3B-II and increased SQSTM1 accumulation in EPCs compared with those in control group. These effects were more pronounced in EPCs from HFD16w mice. In addition, we screened the major mitophagic proteins on

the mitochondrial membrane in EPCs of atherosclerotic mice (Fig. 2e), where both the accumulation of PINK1 and recruitment of PARK2 were decreased (Fig. 2d, f). However, neither BNIP3L/ NIX or MFN2 levels were significantly different compared to those in control mice (Fig. 2g, h).

To further corroborate these findings, we utilized a pH-sensitive tandem GFP-mRFP-LC3 adenovirus to infect EPCs for

**Fig. 1 EPC proliferation inhibition and mitochondrial dysfunction in atherosclerotic mice. a** $ApoE^{-/-}$ mice were fed with HFD and CCK-8 was employed to measure EPC proliferation. CCK-8 results showed that EPC proliferation decreased 25.24% at 8 weeks and 47.31% at 16 weeks respectively in comparison to that of ND. **b** EPCs from ND and atherosclerotic mice were seeded on E-plates and the proliferation ability was real-timely recorded. The normalized cell index indicated EPC proliferation from atherosclerotic mice decreased compared with those from ND mice. Representative graphs were shown from 3 independent experiments. **c** Mitochondrial membrane potential was measured by JC-1. Quantitative analysis of red and green fluorescence showed that the ratio of red related to green fluorescence decreased to 1.204 in HFD8w group ($p < 0.01$) and to 0.458 in HFD16w group ($p < 0.01$). **d** Mitochondrial ROS production was evaluated by MitoSOX. Quantitative analysis indicated that red fluorescence increased in HFD8w group ($p < 0.01$) and this was more pronounced in HFD16w group ($p < 0.01$). **e** EPCs were incubated with JC-1, a fluorescent dye of mitochondrial membrane, for 25 min. Fluorescence was visualized via LCSM. Representative images showed the fluorescence intensity in different groups, scale bar: 50 μm. **f** Mitochondrial superoxide levels were labeled with MitoSOX fluorescence indicator for 20 min. Fluorescence was visualized via LCSM. Representative images showed the fluorescence intensity in different groups, scale bar: 50 μm. **g** Mitochondrial morphology was captured by transmission electron microscope. Representative images showed the mitochondrial morphology in different groups, scale bar: 1 μm. (Cells were isolated from 3 mice for 1 experiment and 5 independent experiments were performed, mean ± SD, **$P < 0.01$).

24 h before observation using laser scanning confocal microscopy (LSCM). Yellow puncta, which were a combination of RFP and GFP fluorescence, represented autophagosomes, whereas free red puncta represented autolysosomes, where acidic pH quenches GFP fluorescence. The results showed that both the yellow and free red puncta decreased significantly in EPCs of HDF8w and HFD16w mice compared with those in ND mice (Fig. 2i), suggesting that the numbers of both autophagosomes and autolysosomes decreased. In addition, we as well employed mtKeima, a useful tool in the assessment of mitophagy level. As shown in Fig. 2j, EPCs from HDF8w and HFD16w mice showed a significant lower mitophagy index in comparison to those from ND mice. Both the LSCM and western blots data confirmed that mitophagy was impaired in EPCs of atherosclerotic mice and that the PINK1/PARK2 pathway might be associated with this process.

**PTV reverses mitophagy and improves proliferation of atherosclerotic mice EPCs.** RTCA and CCK-8 were both used to evaluate the proliferation of EPCs from atherosclerotic mice after exposure to 0, 0.1, 0.5, and 1.0 μM PTV. RTCA results showed that the normalized cell index of EPCs significantly increased with the increase of PTV concentration, indicating that PTV increased proliferative ability in a dose-dependent manner. In accordance with RTCA results, CCK-8 assay revealed similar results after different time intervals and doses of PTV treatments (Fig. 3a, b). To check whether mitophagy was associated with this effect, we measured the turnover of MAP1LC3B-II and accumulation of SQSTM1 as well as the number of autophagosomes and autolysosomes (labeled with GFP and mRFP) in LSCM. PTV reversed the decrease in MAP1LC3B-II as well as the accumulation of SQSTM1 elicited by HFD in EPCs from atherosclerotic mice (Fig. 3c, d). To assess the mitophagosome–lysosome fusion, we treated cells with PTV in combination with bafilomycin A1 (BAFA1), a lysosomal inhibitor. BAFA1 treatment augmented MAP1LC3B-II and the accumulation of SQSTM1 in PTV treatment cells was as well higher than that in cells without PTV (Supplementary Fig. 4). This indicated that the increased MAP1LC3B-II level in PTV treatment group is not because of reduced autophagosome turnover, but increased autophagic flux. We next produced EPCs stably expressing mtKeima. The numbers of autophagosomes (yellow puncta) and autolysosomes (free red puncta) also increased after 0.5 μM PTV treatment for 24 h (Fig. 3e). Pretreatment with autophagy inhibitor, BAFA1 before PTV increased more yellow puncta but decreased free red puncta in merged images, further indicating the activation of autophagy by PTV and the successful blocking of autophagy flux by BAFA1. Moreover, mtKeima assay as well indicated that 0.5 μM PTV for 24 h increased mitophagy index (Fig. 3f). In addition, we labeled mitochondria with Mitotracker Deep Red and visualized

MAP1LC3B with immunofluorescence. Pearson's coefficient analysis revealed that the number of mitochondria colocalized with endogenous MAP1LC3B increased after 0.5 μM PTV treatment for 24 h (Fig. 3g), which represented mitophagosome accumulation. Both experiments indicated that PTV improves atherosclerotic EPCs proliferation along with activating mitophagy.

**Mitophagy inhibition blocks PTV-induced EPC proliferation improvement in atherosclerotic mice.** To address the effects of mitophagy on PTV-induced EPC proliferation amelioration, we utilized gene-silencing as well as pharmacological techniques to inhibit mitophagy activity. *Atg7* was significantly knocked-down 72 h after lentiviral infection (Fig. 4a). Reduction in MAP1LC3B-II transformation and ATG12-ATG5 conjugation, as well as accumulation of SQSTM1 in the Atg7-silenced group, further confirmed that mitophagy was effectively inhibited (Fig. 4a). Next, we applied 0.5 μM PTV to *Atg7*-silenced EPCs for 24 h. The results suggested that inhibition attenuated PTV-induced EPC proliferation improvement compared to that in the PTV alone group (Fig. 4b). A similar pattern emerged after application of 3-methyladenine (3-MA), an autophagy pharmacological inhibitor, before PTV treatment in EPCs of atherosclerotic mice (Fig. 4c). Both datasets demonstrated that PTV-induced mitophagy contributes to EPCs proliferation improvement.

**PTV induces mitophagy in EPCs from atherosclerotic mice via PINK1-PARK2 pathway.** We detected reduced expression of PINK1 and PARK2 in EPCs from atherosclerotic mice. Thus, to examine whether this was related to PTV-activated mitophagy, we analyzed PINK1 accumulation and PARK2 recruitment in the mitochondria of EPCs. The western blots results suggested that PTV increased PINK1 accumulation and PARK2 recruitment in mitochondrial membrane in a dose-dependent manner (Fig. 4d–f). This was confirmed by immunofluorescence analysis under LSCM. PARK2 was translocated from the cytoplasm to localize at the mitochondrial membrane after 0.5 μM PTV treatment for 24 h (Fig. 4g). Moreover, autophagic marker MAP1LC3B dots were extensively colocalized with PARK2 under PTV treatment, indicating that PINK1-PARK2-mediated mitophagy served as the major effect in EPCs (Fig. 4h).

To further corroborate these findings, we used shRNA to knock down *Pink1* and *Park2*, respectively (Fig. 5a). Silencing *Pink1* or *Park2* reduced the PTV-mediated increase in MAP1LC3B-II levels (Fig. 5b–e). Additionally, silencing *Pink1* or *Park2* reversed the proliferation improvement of PTV in CCK-8 assays (Fig. 5f, g). A similar pattern was shown in the RTCA experiment (Fig. 5h, i). For removing the interference of residual PINK1 on mitophagy completely, we established *Pink1* KO mice.

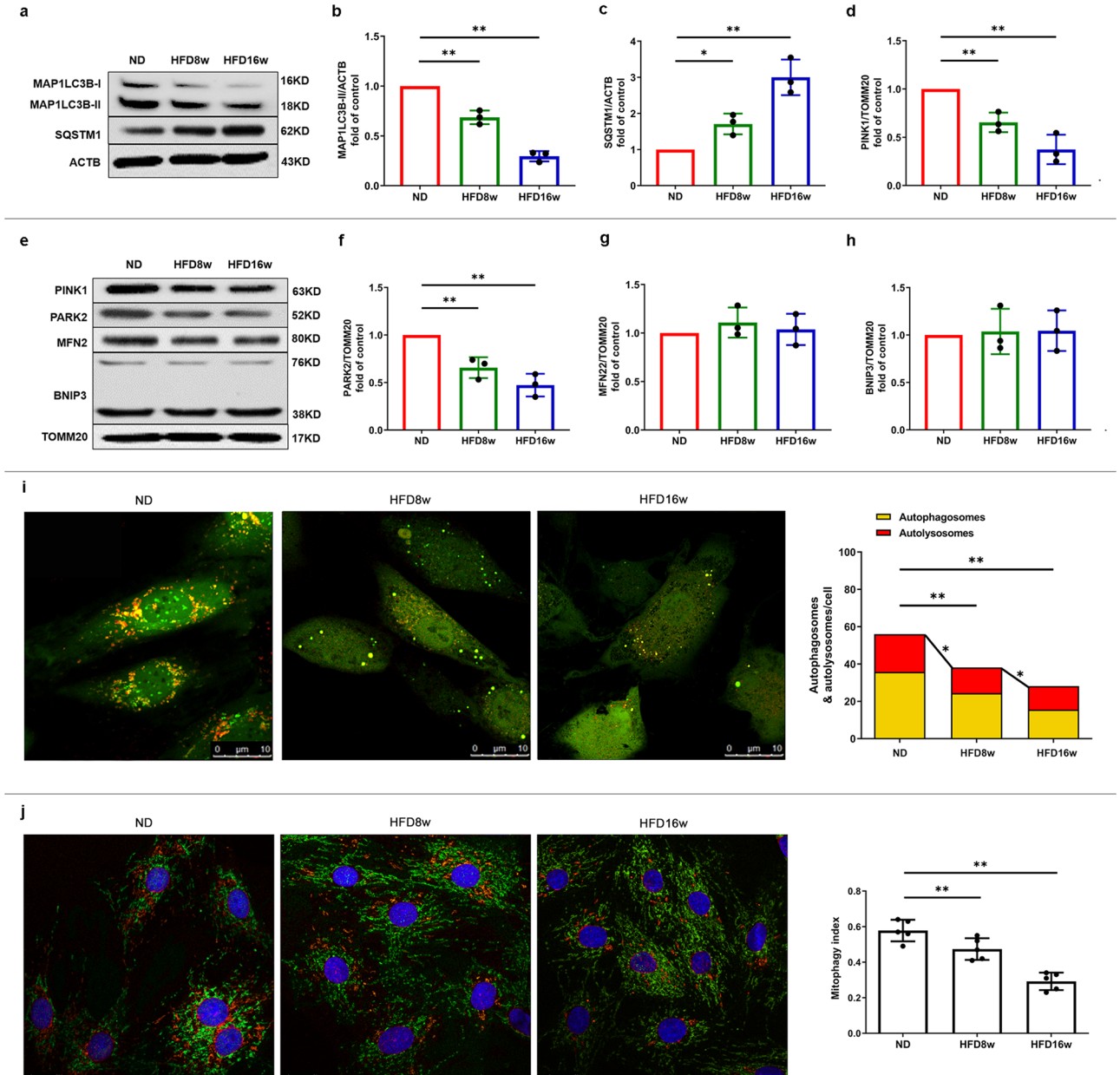

**Fig. 2 Mitophagy was impaired in atherosclerotic mice. a** Western blots revealed that MAP1LC3B-II turnover decreased (**b**) and the expression of SQSTM1 increased (**c**) significantly in atherosclerotic mice. **e** Representative western blots of mitophagic proteins in EPCs of atherosclerotic mice. (**d**, **f**) Quantitative analysis indicated that both the expression of PINK1 and PARK2 decreased on the mitochondrial membrane in EPCs of atherosclerotic mice. (**g**, **h**) Quantitative analysis revealed that MFN and BNIP3 showed no markedly difference in EPCs between ND and atherosclerotic mice. **i** Representative images and quantitative analysis of yellow and free red puncta in merged images in EPCs from ND and atherosclerotic mice, scale bar: 10 μm. **j** EPCs from ND and atherosclerotic mice were transfected by mtKeima for 12 h. EPCs from HFD8w and HFD16w *ApoE*$^{-/-}$ mice showed a significant lower mitophagy index in comparison to those from ND mice. Scale bar: 25 μm. (EPCs were isolated from ND, HFD8w and HFD16w *ApoE*$^{-/-}$ mice, $n = 10$ cells per group, cells were isolated from 3 mice for 1 experiment, and 3 independent experiments were performed, mean ± SD, *$P < 0.05$, **$P < 0.01$).

As is shown in Fig. 5j, negative signal of PINK1 and less expression of PARK2 in *Pink1* KO mice represented the stable CRISPR technique for removing sequence. As expected, EPCs isolated from *Pink1* KO mice showed less MAP1LC3B (Supplementary Fig. 5a) and decreased co-localization with mitochondria as compared to the WT mice (Supplementary Fig. 5b, c). Consistent with sh*Pink1* group, similar decreased EPC proliferative ability were also observed in *Pink1* KO mice (Fig. 5k, l). Both impairment of mitophagy and EPC proliferation caused by PINK1 ablation can be reversed by PTV treatment. That all strongly indicated that the presence of PINK1 and downstream

PINK1-PARK2 pathway are essential for PTV-activated mitophagy, which contributes to maintain the normal EPC proliferation.

**PTV elicits mitochondrial calcium release.** As it is still unclear whether PTV is capable of activating the calcium signal pathway in EPCs and whether the change in $[Ca^{2+}]_i$ or $[Ca^{2+}]_m$ is associated with mitophagy induction, we treated EPCs with various concentrations of PTV for 24 h and the calcium probe fluo3-AM. PTV increased fluorescence intensity in EPCs under an LSCM.

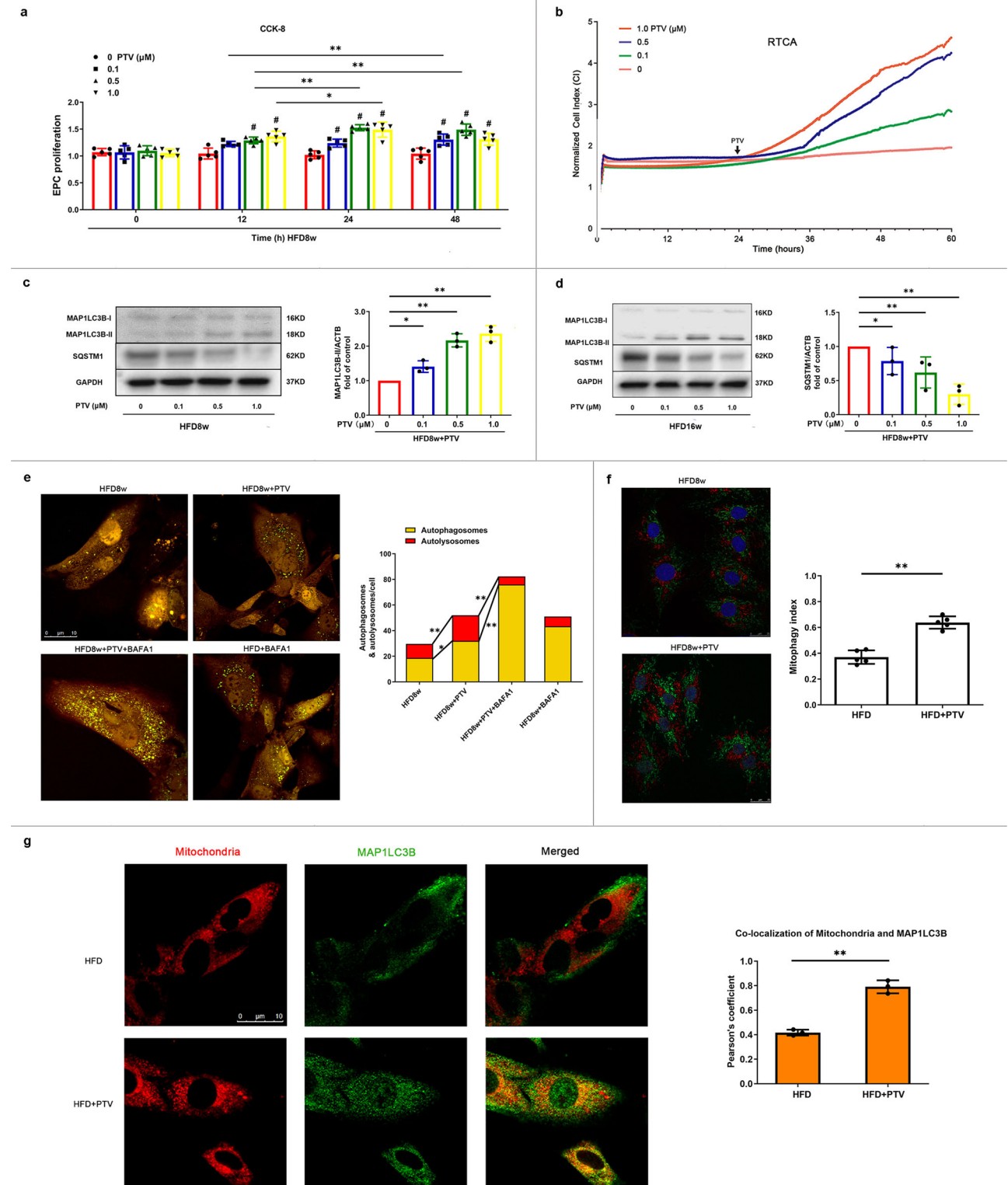

The higher the PTV concentration, the stronger the fluorescence intensity was. $F_{min}$ was detected by measuring fluorescence intensity in the presence of EGTA (extracellular $Ca^{2+}$ chelator) and BAPTA-AM (cell-permeable intracellular $Ca^{2+}$ chelator). $F_{max}$ was obtained by saturating intracellular $Ca^{2+}$ (ionomycin plus $Ca^{2+}$). Calculation of $[Ca^{2+}]_i$ (Methods) revealed that PTV significantly increased $[Ca^{2+}]_i$ (Fig. 6a, b). To further elucidate where the $[Ca^{2+}]_i$ derived from after PTV treatment, we marked $[Ca^{2+}]_i$ with fluo3-AM and $[Ca^{2+}]_m$ with Rhod2-AM in

$Ca^{2+}$ free medium, separately. Then we added various concentrations of PTV to $Ca^{2+}$ free medium under LSCM. PTV induced a $[Ca^{2+}]_i$ transient increase especially 0.5 μM (Fig. 6c, green line) and 1.0 μM (Fig. 6c, red line). In rhod2-AM marked $Ca^{2+}$ free medium, we detected a transient $[Ca^{2+}]_m$ dose-dependent decrease (Fig. 6d, particularly at PTV 0.5 μM [green line] and 1.0 μM [red line]). These results strongly suggest that $[Ca^{2+}]_i$ increase is in part derived from mitochondrial calcium release.

**Fig. 3 PTV improved EPC proliferation and reversed impaired mitophagy in atherosclerotic mice. a** CCK-8 results showed that PTV improved atherosclerotic mice EPC proliferative activity in dose- and time-dependent manners after 0, 12, 24, or 48 h PTV exposure at a series of concentrations (0, 0.1, 0.5 or 1.0 μM). **b** After seeding on E-plates for 24 h, EPCs were treated with different concentrations of PTV (0, 0.1, 0.5, or 1.0 μM) respectively (black arrow) and monitored by RTCA. The normalized cell index indicated PTV dose-dependently decreased EPC proliferation. **c** HFD8w $ApoE^{-/-}$ mice EPCs were incubated with PTV (0, 0.1, 0.5 or 1.0 μM) for 24 h. Western blots revealed that PTV markedly increased the turnover of MAP1LC3B-II and reduced expression of SQSTM1 in dose-dependent manners. **d** HFD16w $ApoE^{-/-}$ mice EPCs were incubated with PTV (0, 0.1, 0.5 or 1.0 μM) for 24 h. Western blots revealed that PTV markedly increased the turnover of MAP1LC3B-II and reduced expression of SQSTM1 in dose-dependent manners. **e** EPCs from atherosclerotic mice were infected by tandem GFP-mRFP-LC3 adenovirus for 24 h before exposure to PTV (0.5 μM) 24 h, BAFA1 (10 nM) 6 h plus ox-LDL (0.5 μM) 24 h or BAFA1 (10 nM) 6 h alone. Representative images and quantitative analysis of yellow and free red puncta formation in different groups. Scale bar: 10 μm. **f** EPCs from atherosclerotic mice were transfected by mtKeima plasmid for 12 h before exposure to PTV (0.5 μM) 24 h. Statistical analysis showed that PTV increased mitophagy index, indicating that PTV increased mitophagy in EPCs. Scale bar: 25 μm. **g** Atherosclerotic mice EPCs were incubated with 0.5 μM PTV for 24 h. Mitotracker Deep Red was used to mark mitochondria and immunofluorescent staining was applied to mark MAP1LC3B in atherosclerotic EPCs. Representative merged images and Pearson's overlap coefficient analysis showed that the yellow area increased in PTV treatment group compared with HFD group, scale bar: 10 μm. (*n* = 10 cells per group, EPCs were isolated from $ApoE^{-/-}$ mice fed with high-fat diet for 8 weeks, cells were isolated from 3 mice for 1 experiment and 3 independent experiments were performed, mean ± SD, *$P$ < 0.05, **$P$ < 0.01).

**CAMK1 activation contributes to PTV-induced mitophagy.** Intracellular calcium was the major regulator of mitophagy and the downstream protein kinases were reported to induce mitophagy or autophagy[30,31]. Whether PTV-induced $[Ca^{2+}]_i$ increase is associated with mitophagy activation as well is unknown. Therefore, we accordingly measured the activity of calcium-dependent protein kinases. We measured the activity of calcium-dependent protein kinases by screening the phosphorylation status of the CAMK family after PTV treatment. The data indicated that PTV increased phosphorylation of CAMK1 at the Thr[177] site in EPCs in a dose-dependent manner (Fig. 6e).

To address the impact of activation of CAMK1 on mitophagy induction, we applied BAPTA-AM (20 μM), a cell-permeable intracellular $Ca^{2+}$ chelator, to pretreat EPCs before PTV exposure. BAPTA-AM effectively inhibited phosphorylation of CAMK1 at Thr[177] site (Fig. 7a, c), meanwhile reversed PTV-induced mitophagy (Fig. 7b, d). We also utilized gene-silencing techniques to knockdown *Camk1* (Fig. 7e), which reduced the turnover of MAP1LC3B-II (Fig. 7f, g), the number of autophagosomes (Fig. 7h) and mitophagy index (Fig. 7i) increased by PTV. Furthermore, knockdown *Camk1* decreased the colocalization of MAP1LC3B with mitochondria in PTV treatment (Fig. 7j), implicating that CAMK1 was related to PTV-activated mitophagy.

**CAMK1 contributes to phosphorylation of PINK1 to induce mitophagy.** In this study, we have confirmed that both the CAMK1 and the canonical PINK1-PARK2 pathway are associated with mitophagy activation after PTV treatment. Consequently, we explored whether CAMK1 was associated with PINK1-PARK2 pathway. As the activity of PINK1-PARK2-mediated mitophagy depends on the phosphorylation of Ser[228] in PINK1 and Ser[65] in PARK2[32,33], we assessed the serine phosphorylation of PINK1 and PARK2 in different treatment groups. PTV pronouncedly increased the ratio of Ser[228] phosphorylation in PINK1 in comparison to that in the control groups (Fig. 8a, b). Similarly, PTV up-regulated the ratio of Ser[65] phosphorylation in PARK2 (Fig. 8a–c). While both the phosphorylation of PINK1 and PARK2 were significantly decreased by using siRNA targeting the *Camk1*. It has been reported that the expression and phosphorylation of CAMK1 is regulated in response to $Ca^{2+}$ signaling, which is the consequence of the increased intracellular $Ca^{2+}$ concentration rather than changes of osmotic stress[34]. Thus, we used a high-calcium medium to induce the CAMK1 phosphorylation by elevated intracellular $Ca^{2+}$ concentration. Compared to the control group, the proportion of phosphorylated PINK1 was up-regulated by high calcium-mediated CAMK1 activation. This effect was significantly

reversed when knockdown *Camk1*, which revealed the essential role of activated CAMK1 in PINK1 phosphorylation (Supplementary Fig. 6a). PINK1, known as the upstream serine/threonine kinase, is also regulated by serine phosphorylation[32]. We next employed an in vitro kinase assay to observe whether CAMK1 is associated with PINK1 phosphorylation independent of other proteins. As shown in Supplementary Fig. 6b, in vitro kinase assay further demonstrated that CAMKI contributed to PINK1 phosphorylation on serine[228]. Thus, our above data supported that CAMK1 is closely associated with PINK1 phosphorylation and PTV-induced mitophagy. However, whether CAMK1 directly and/or indirectly phosphorylates PINK1 still needs further explore.

**Mitophagy-dependent homeostasis maintenance and ROS clearance contribute to PTV beneficial effects on EPCs.** PTV treatment alleviated ROS accumulation in the mitochondria and reversed mitochondrial swelling and cristae fracture in atherosclerotic EPCs (Fig. 8d). When mitophagy was inhibited by knockdown of *Atg7* or *Pink1*, the effects of PTV on ROS clearance were decreased (Fig. 8e, g). In addition, both in *Atg7* + PTV and *Pink1* + PTV treatment groups, impairment mitochondria accumulated, mitochondria swelled and cristae fractured in EPCs (Fig. 8d), indicating that PTV-induced mitophagy contributed to mitochondrial homeostasis maintenance. To confirm CAMK1 was associated with this process, we silenced *Camk1* by LV transfection before PTV application. We found that the beneficial effects of PTV on mitochondria in atherosclerotic EPCs were reversed in *Camk1* silenced groups. Unsurprisingly, compared with the PTV treatment EPCs, decreased MMP (Fig. 8e, g) and accumulation of ROS (Fig. 8f, h) was shown in the sh*Pink1* + PTV, sh*Camk1* + PTV and *Atg7* + PTV groups. These results suggested that CAMK1-PINK1-mediated mitophagy was associated with PTV-involved ROS clearance and mitochondrial homeostasis maintenance in atherosclerotic EPCs.

**Disrupting of CAMK1-meidated mitophagy prevents EPC-mediated repairment of vascular endothelium.** To investigate whether CAMK1-mediated EPC mitophagy contributed to repairment of damaged vessels, we established a model of carotid artery intima injury in mice. AcLDL-DiI labeled pEGFP-N2-EPCs were pre-treated by 0.5 μm PTV for 24 h. All these mice have received EPC transplantation treatment. After 7 days of transplantation, labeled cells were traced to home and showed red (acLDL-DiI) and green fluorescence in line in carotid artery intima (Fig. 9a, white arrow). However, silencing *Atg7* or *Camk1* before transplantation markedly reduced EPC proliferation and

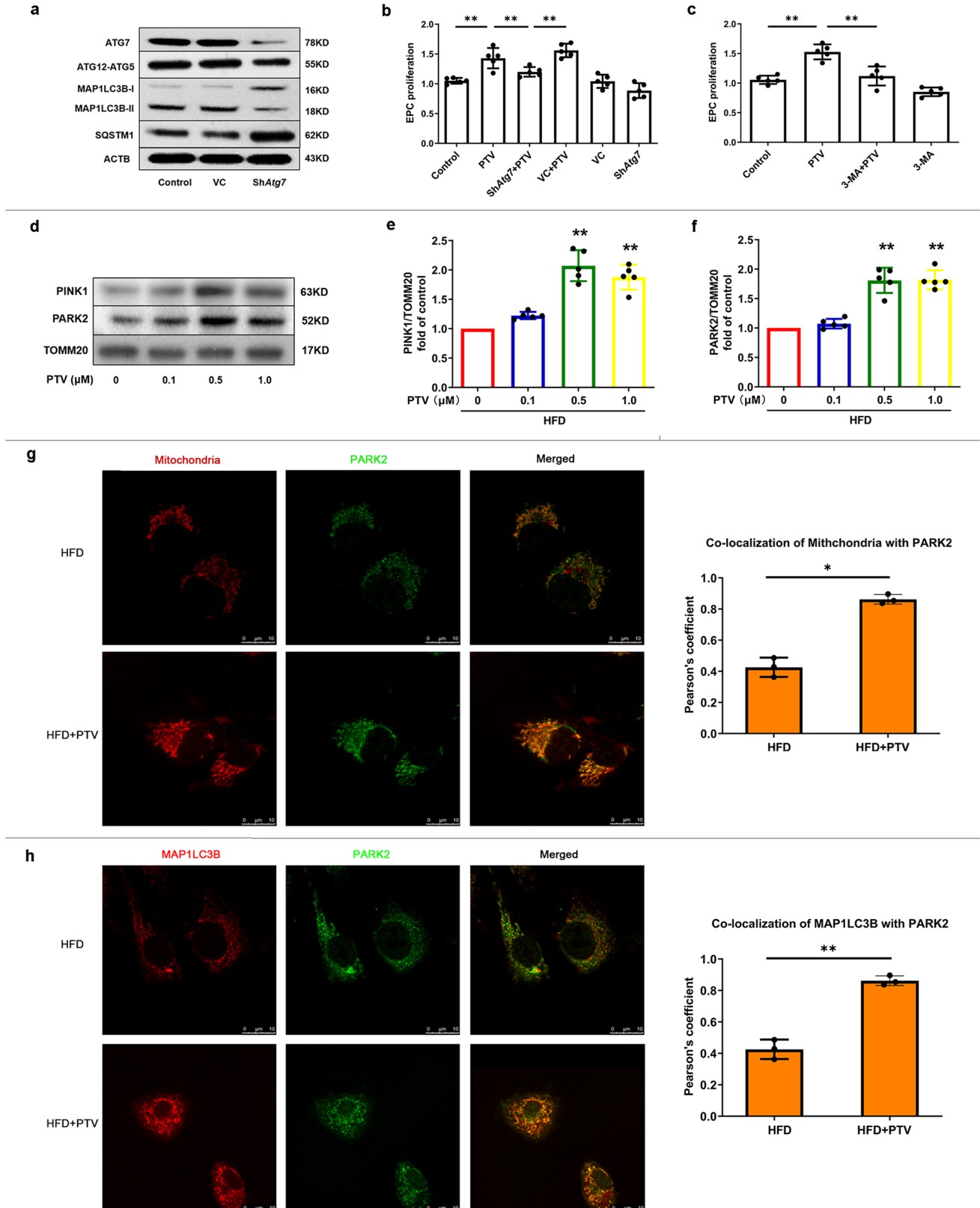

home to carotid artery intima in comparison to PTV and PTV + VC groups (Fig. 9a, white arrow). Furthermore, Evans Blue staining was performed to evaluate the reendothelialized area after EPC transplantation in mice. As shown in Fig. 9b, the reendothelialized area in PTV group was significantly larger than in the PTV + sh*Atg7* and PTV + sh*Camk1* groups. These results demonstrated that either knockdown *Atg7* or *Camk1* significantly reduced EPC-mediated reendothelialization of injured carotid arteries, indicating CAMK1-mediated mitophagy is required for EPC function and repairment of damaged vessels.

**Fig. 4 PTV activated PINK1-PARK2 dependent mitophagy. a** Representative western blots for the detection of ATG7, ATG12–ATG5 conjugate, MAP1LC3B-II, and SQSTM1 after infection showed that *Atg7* was successfully knocked down and mitophagy was effectively inhibited in *Atg7* silencing group. **b** Silencing *Atg7* before PTV treatment significantly reduced proliferative activity compared with PTV alone group. **c** 3-MA (2 mM) was added to inhibit mitophagy before 0.5 μM PTV treatment. CCK-8 assay showed that the proliferative activity in 3-MA + PTV group reduced significantly compared with PTV alone group. (**d–f**) PTV treatment (0, 0.1, 0.5, or 1.0 μM) for 24 h increased PINK1 accumulation and PARK2 recruitment in mitochondrial membrane in dose-dependent manner. **g** Mitotracker Deep Red was used to mark mitochondria and immunostain was employed to mark PARK2 in atherosclerotic mice EPCs. Merged images and Pearson's overlap coefficient analysis indicated that PARK2 mostly localized on mitochondria in PTV treatment EPCs related to HFD group, scale bar: 10 μm. **h** Atherosclerotic mice EPCs were co-immunostained for MAP1LC3B and PARK2. Overlap of PARK2 with MAP1LC3B were observed in PTV treatment EPCs in comparison to HFD group and Pearson's overlap coefficient was employed to analyze the co-localization, scale bar: 10 μm. (n = 10 cells per group, EPCs were isolated from *ApoE*−/− mice fed with high-fat diet for 8 weeks, cells were isolated from 3 mice for 1 experiment and 3 independent experiments were performed, mean ± SD, *P < 0.05, **P < 0.01).

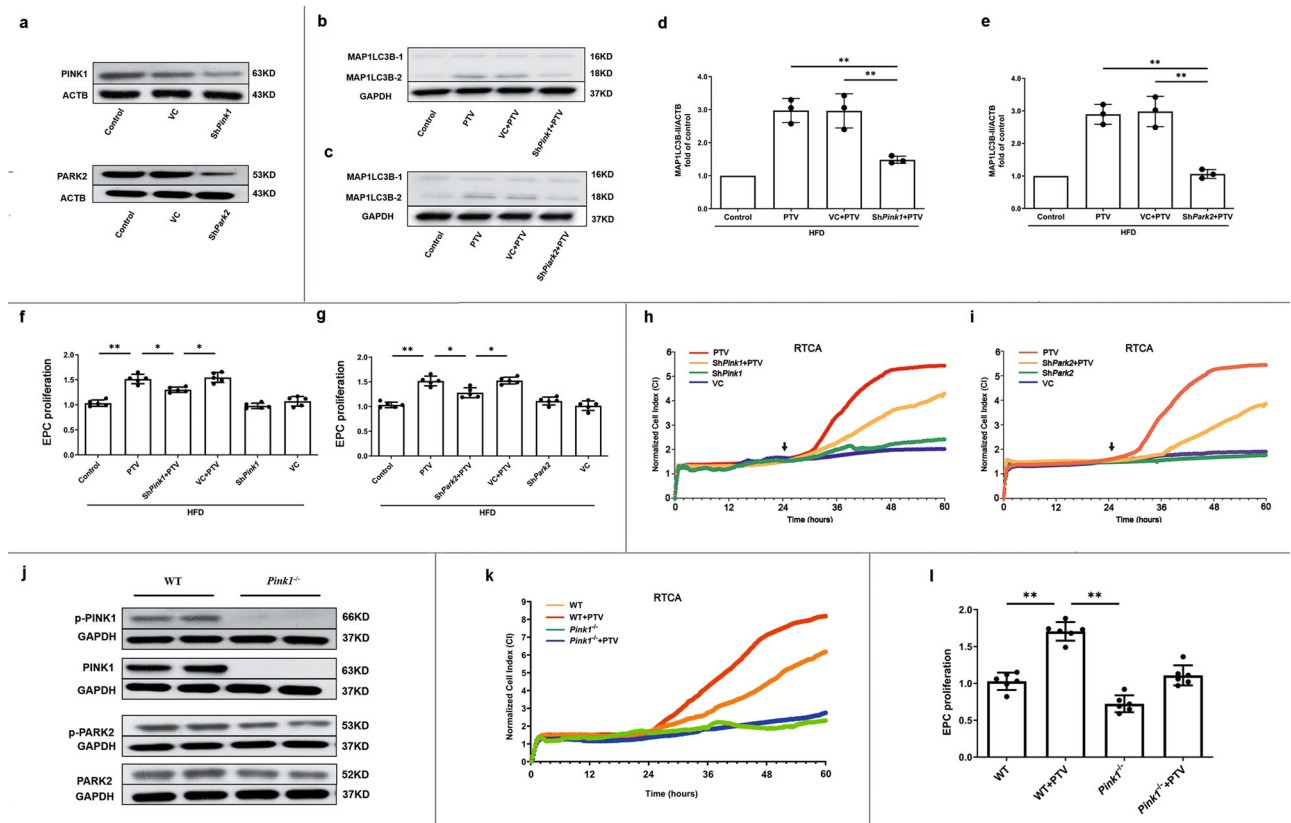

**Fig. 5 PINK1-PARK2 pathway contributed to EPC proliferation. a** Representative western blots showed that shRNA targeting *Pink1* (sh*Pink1*) effectively silenced PINK1 protein expression and shRNA targeting *Park2* (sh*Park2*) effectively silenced PARK2 protein expression after 24 h infection, respectively. Representative western blots and quantitative analysis revealed that either silencing *Pink1* (**b**, **d**) or *Park2* (**c**, **e**) significantly reduced MAP1LC3B expression. CCK-8 assay showed that either silencing *Pink1* (**f**) or *Park2* (**g**) before PTV treatment significantly reduced proliferative activity compared with PTV alone group. **h** EPCs of silencing *Pink1* were seeded on E-plates for 24 h, 0.5 μM PTV was added to E-plates in PTV and sh*Pink1* + PTV groups after 24 h incubation (black arrow). The normalized cell index indicated that EPC proliferative activity decreased in sh*Pink1* + PTV group compared with PTV alone group. **i** EPCs of silencing *Park2* were seeded on E-plates for 24 h, 0.5 μM PTV was added to E-plates in PTV and sh*Park2* + PTV groups after 24 h incubation (black arrow). The normalized cell index indicated that EPC proliferative activity decreased in sh*Park2* + PTV group compared with PTV alone group. **j** Western blots of the expression of PINK1, PARK2 and their phosphorylated form in WT and *Pink1*−/− mice. EPCs with or without 0.5 μM PTV treatment were seeded on E-plates for 24 h, 0.5 μM in PTV and *Pink1*−/− + PTV groups after 24 h incubation. The normalized cell index (**k**) and CCK-8 (**l**) indicated that EPC proliferative activity fell down in WT and *Pink1*−/− mice with or without PTV treatment. (EPCs were isolated from *ApoE*−/− mice fed with high-fat diet for 8 weeks, cells were isolated from 3 mice for 1 experiment and 3 independent experiments were performed, mean ± SD, *P < 0.05, **P < 0.01).

## Discussion

In the current study, we revealed both proliferation inhibition and mitophagy impairment in atherosclerotic EPCs. We showed that PTV elicits calcium release from mitochondria to activate CAMK1, which increases the level of phosphorylated PINK1 and PINK1 further recruits PARK2; PARK2 then localizes to the mitochondrial membrane and was phosphorylated to activate mitophagy (Fig. 9c). Moreover, CAMK1-PINK1-mediated

mitophagy maintains EPC proliferation under atherosclerotic conditions, depending on ROS clearance and maintenance of mitochondrial homeostasis.

Mitochondria are an abundant source of energy in most cell types. Mitochondrial homeostasis depends on mitochondrial dynamics, biogenesis, and the timely removal of worn-out portions. Mitophagy specifically eliminates damaged and DNA mutant mitochondria, and exerts a major role in homeostatic

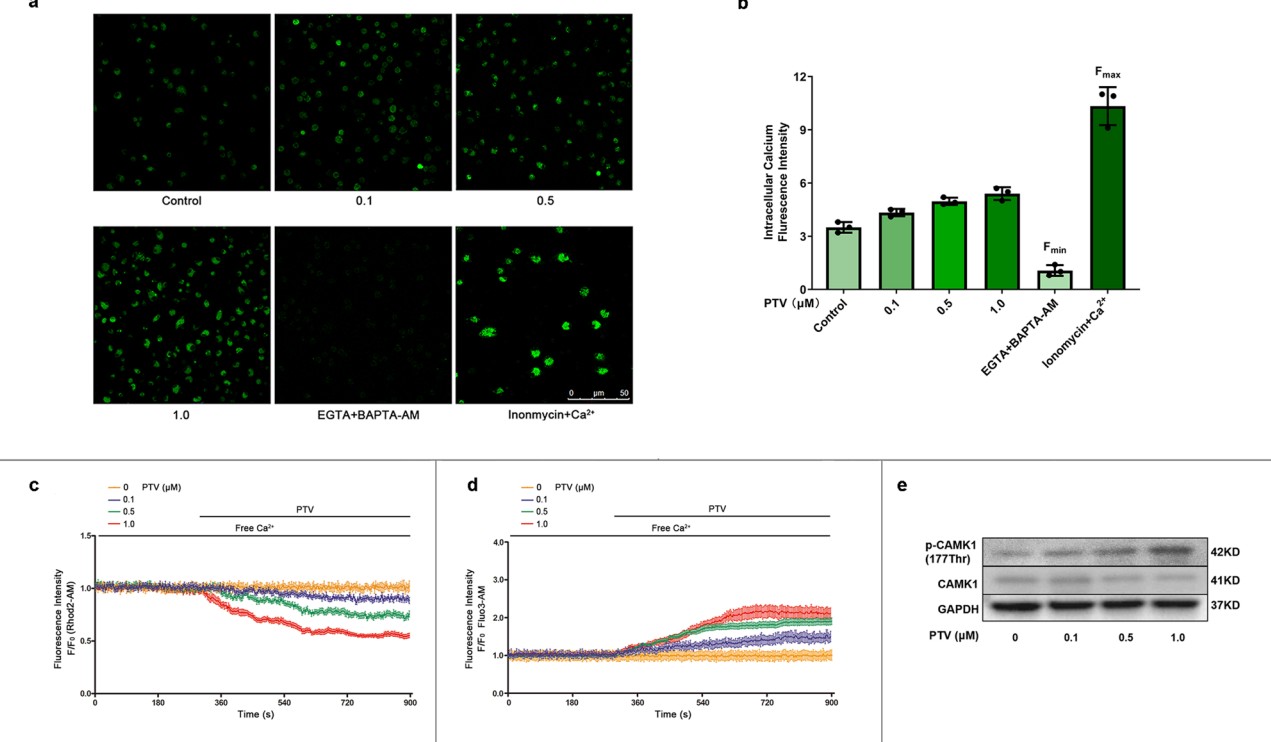

**Fig. 6 PTV elicited mitochondrial calcium release and activated CAMK1. a** EPCs were incubated with different concentrations of PTV (0, 0.1, 0.5 or 1.0 μM) for 24 h. Then cells were incubated with calcium probe fluo3-AM. The representative images showed the fluorescence intensity in different groups. EGTA + BAPTA-AM was used to obtain the minimum fluorescence intensity ($F_{min}$) while inonmycin+calcium was used to obtain the maximum fluorescence intensity ($F_{max}$), scale bar: 50 μm. **b** Quantitative analysis of fluorescence intensity showed that PTV treatment markedly increased intracellular calcium concentration. **c** Mitochondrial calcium was marked by Rhod2-AM probe. Different concentrations of PTV (0, 0.1, 0.5 or 1.0 μM) were added to the free calcium medium. PTV elicited mitochondrial calcium release in dose-dependent manner. **d** Intracellular calcium was marked by Fluo3-AM probe. Different concentrations of PTV (0, 0.1, 0.5 or 1.0 μM) were added to the free calcium medium. PTV elicited intracellular calcium increase in dose-dependent manner. **e** Representative western blots for phosphorylation and total CAMK1 after 0.5 μM PTV treatment for 24 h indicated that PTV dose-dependently increased phosphorylation of CAMK1 at Thr[177] site. (EPCs were isolated from $ApoE^{-/-}$ mice fed with high-fat diet for 8 weeks, cells were isolated from 3 mice for 1 experiment and 3 independent experiments were performed, mean ± SD, *$P < 0.05$, **$P < 0.01$).

quality control, as well as contributing to physiological processes, such as erythrocyte maturation[35,36] and α-synuclein degradation in neurons[37,38]. MtDNA damage and deletion lead to reduced complex I activity and decreased oxidative phosphorylation, which further increased mitochondrial dysfunction. The accumulation of damaged mitochondria further results in ROS overproduction, cellular disorder, and even apoptosis[39]. In the present study, EPC mitophagy was impaired in the atherosclerotic environment, which resulted in ROS accumulation and EPC proliferation inhibition. On the contrary, the activation of mitophagy ameliorates mitochondrial dysfunction and cell toxicity in disease. In Alzheimer's disease, mitophagy induction degrades amyloid-beta (Aβ) and reduces plaque formation and memory deficits in animal models[40,41]. In diabetes mellitus (DM), mitochondrial dysfunction results from advanced glycation end (AGE) product accumulation aggravated by hyperglycemia-induced glycoxidative stress; mitophagy degrades AGEs and clears damaged mitochondria. Mitophagy activation in platelets alleviates thrombotic injuries in DM[42]. Hence, mitophagy safeguards against disease and represents a promising therapeutic target.

Since mitophagy is a form of selective macroautophagy, they share partial molecular processes and pathways, such as both the formation of autophagosomes and mitophagosomes needs MAP1LC3 cleavage and ubiquitination in phagophore and envelops the substrate to lysosomes[43]. In this study, we employed MAP1LC3B and SQSTM1 to evaluate mitophagy, in addition, we found that mitophagy-related PINK1 and PARK2 were decreased in EPC from atherosclerosis mice and mitophagy flux data confirmed mitophagy inhibition, which suggested that mitophagy was the major mechanism instead of macroautophagy. Importantly, the vast majority of MAP1LC3B colocalized with mitochondria, and MAP1LC3B colocalized with PARK2. It is now generally believed that PINK1 was canonical mitophagy protein, and PINK1, PARK2, and ubiquitin have pivotal roles in priming mitophagy. However, the entire regulatory landscape and the precise control mechanisms of mitophagy remain to be elucidated. We used *Pink1* KO mice and PINK1 silencing method to disturb mitophagy to confirm that the benefit effect of PTV on EPC contributed to mitophagy reversion. Therefore, we considered that PTV protect EPC dependent on PINK1-PARK2 pathway-mediated mitophagy, rather than macroautophagy.

To date, several mitophagy receptor systems have been mechanistically elucidated. The well-characterized PINK1-PARK2 program targets mitophagy receptors to depolarize mitochondria via ubiquitylation of proteins on the OMM. PINK1 senses the mitochondrial depolarization state, accumulates on the OMM, and recruits PARK2 from the cytoplasm. After recruitment, E3 ligase PARK2 ubiquitylates numerous LIR-containing autophagy receptor proteins in OMM, including SQSTM1[19], OPTN[20], and NBR1[44]. These ubiquitylated proteins bind with MAP1LC3 through the LIR motif, which promotes mitochondria engagement and sequestration by autophagosomes. Another mitophagy receptor system is the transmembrane

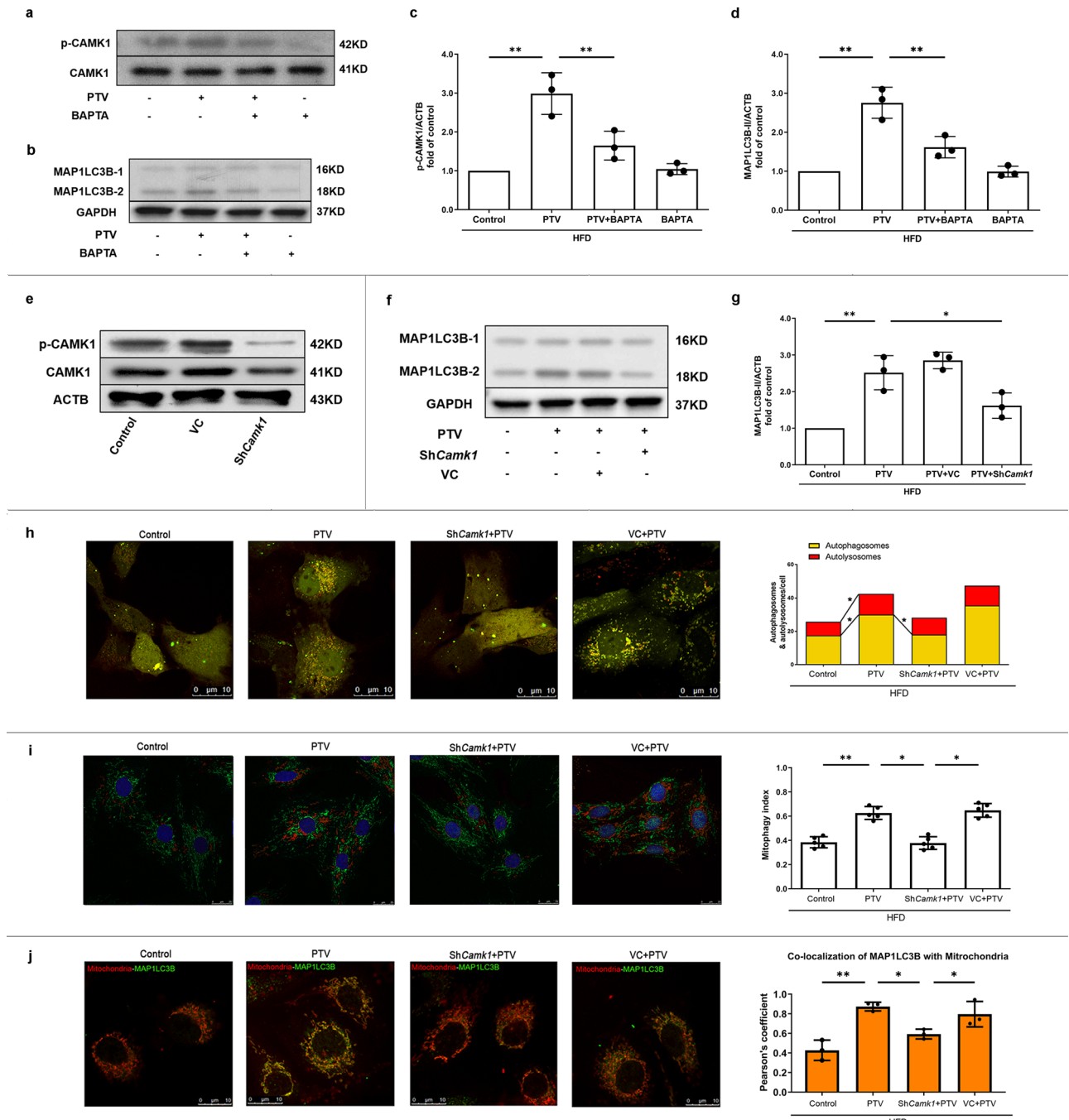

**Fig. 7 CAMK1 activation contributes to PTV-induced mitophagy. (a–d)** EPCs were pretreated with BAPTA-AM (20 mM) for 20 min followed by PTV (0.5 μM) for 24 h. Representative western blots (**a**) and quantitative analysis (**c**) indicated that BAPTA-AM significantly decreased PTV-induced CAMK1 phosphorylation at Thr[177] site. Representative western blots (**b**) and quantitative analysis (**d**) indicated that BAPTA-AM significantly decreased the expression of MAP1LC3B-II induced by PTV. **e** Lentiviral vector carrying *Camk1* shRNA was employed to knockdown *Camk1* in EPCs. Representative western blots showed that *Camk1* was effectively knocked down. *Camk1* was knocked down by shRNA for 72 h before PTV treatment. Representative western blots (**f**) and quantitative analysis (**g**) showed that *Camk1* knockdown significantly decreased the expression of MAP1LC3B-II induced by PTV. **h** EPCs from atherosclerotic mice were transfected by lentivirus to knock down *Camk1*. Camk1 knockdown group, control group, and vector control group EPCs were infected by tandem GFP-mRFP-LC3 adenovirus for 24 h before exposure to PTV (0.5 μM) 24 h. Representative LSCM images showed puncta formation in different groups. Scale bar: 10 μm. Quantitative analysis of yellow and free red puncta. PTV increased the number of yellow and free red puncta compared with control. Camk1 knockdown significantly decreased the number of yellow puncta increased by PTV. **i** EPCs from atherosclerotic mice were transfected by lentivirus to knock down *Camk1*. All the groups were infected by mtKeima plasmid for 12 h before treatment by PTV PTV (0.5 μM) 24 h. Representative images showed puncta formation in different groups. Scale bar: 25 μm. PTV treatment increased the red fluorescence intensity turnover, indicating that more mitochondria were transferred to lysosomes. This effect was blocked by silencing *Camk1*. Quantitative analysis of the fluorescent area showed that PTV increased mitophagy index, but knock down *Camk1* blocked this effect. **j** Merged images revealed that *Camk1* knockdown reduced the co-localization of MAP1LC3B and mitochondria in EPCs according to Pearson's overlap coefficient analysis. Scale bar: 10 μm. ($n = 10$ cells per group, EPCs were isolated from 3 $ApoE^{-/-}$ mice fed with high fat diet for 8 weeks and 3 independent experiments were performed, mean ± SD, *$P < 0.05$, **$P < 0.01$).

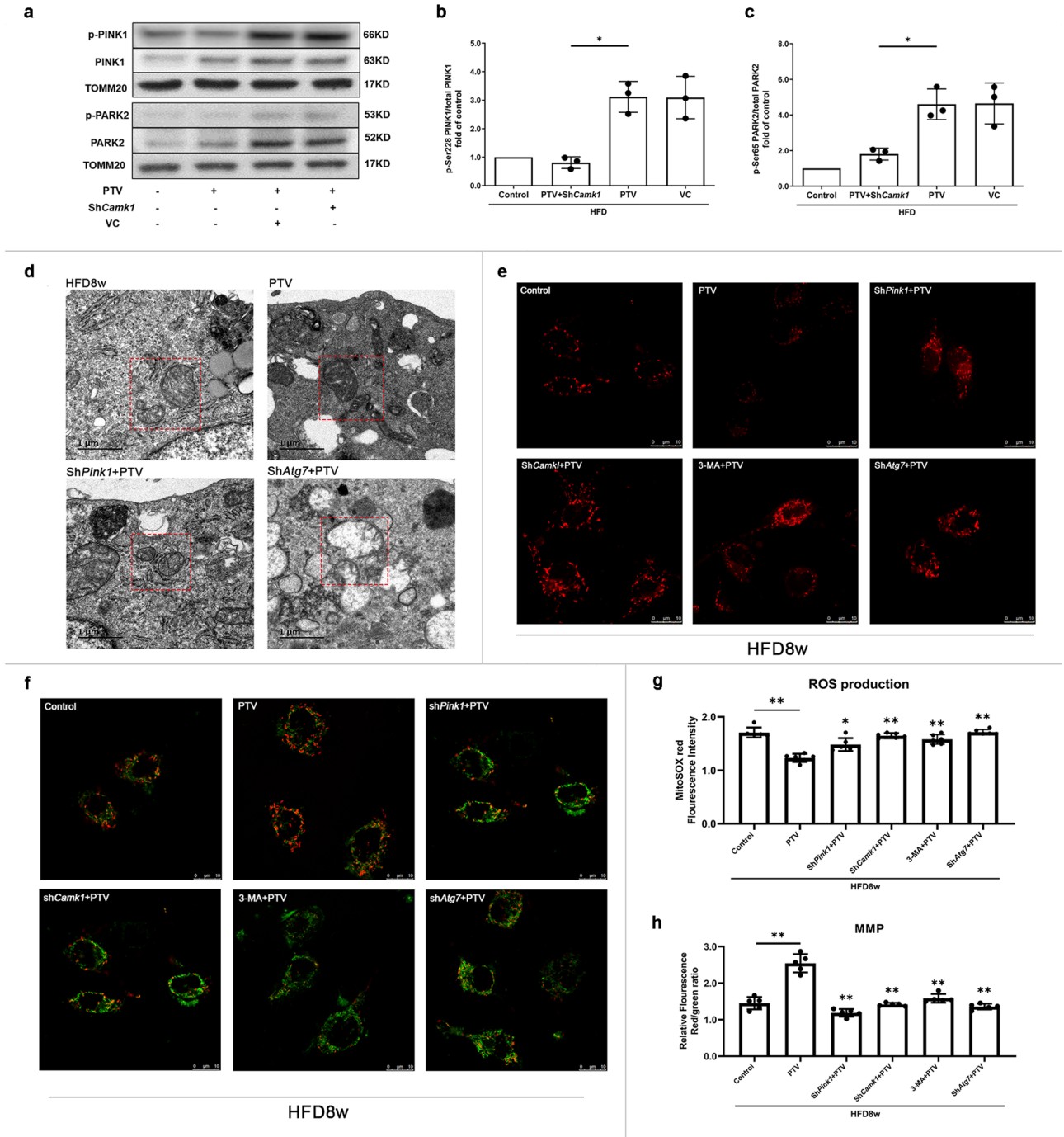

**Fig. 8 CAMK1 contributes to phosphorylation of PINK1 and PARK2 to PTV-induced mitophagy. EPCs were stably transfected with *Camk1* knockdown lentiviral vector 72 h before 0.5 μm PTV treatment for 24 h. a** Immunoblot was employed to detect phosphorylation of PINK1 Ser[228] and PARK2 Ser[65] in each group. We normalized the expression of total PINK1 or PARK2 to compare the phosphorylation level. **b** Quantitative analysis revealed that PTV significantly up-regulated Ser[228] phosphorylation in PINK1 but *Camk1* knockdown remarkably reduced this effect in PECs from atherosclerotic mice. **c** Quantitative analysis revealed that PTV significantly increased PARK2 Ser[65] phosphorylation but *Camk1* knockdown reversed this effect in PECs from atherosclerotic mice. **d** PTV treatment reversed HFD8w-induced mitochondrial swelling and rupture of mitochondrial cristae; either PINK1 or ATG7 silencing blocked the effect of PTV on mitochondria of EPCs. **e** PTV treatment reversed HFD8w-induced ROS production, scale bar: 10 μm. **f** PTV treatment reversed HFD 8w-induced MMP decreased, scale bar: 10 μm. Quantitative analysis indicated the effect of PTV on ROS (**g**) and MMP (**h**) of EPCs can be inhibited by silencing ATG7, PINK1 blocked, and 3-MA pretreatment. (EPCs were isolated from 3 *ApoE*[−/−] mice fed with high fat diet for 8 weeks and 3 independent experiments were performed, mean ± SD, *$P < 0.05$, **$P < 0.01$).

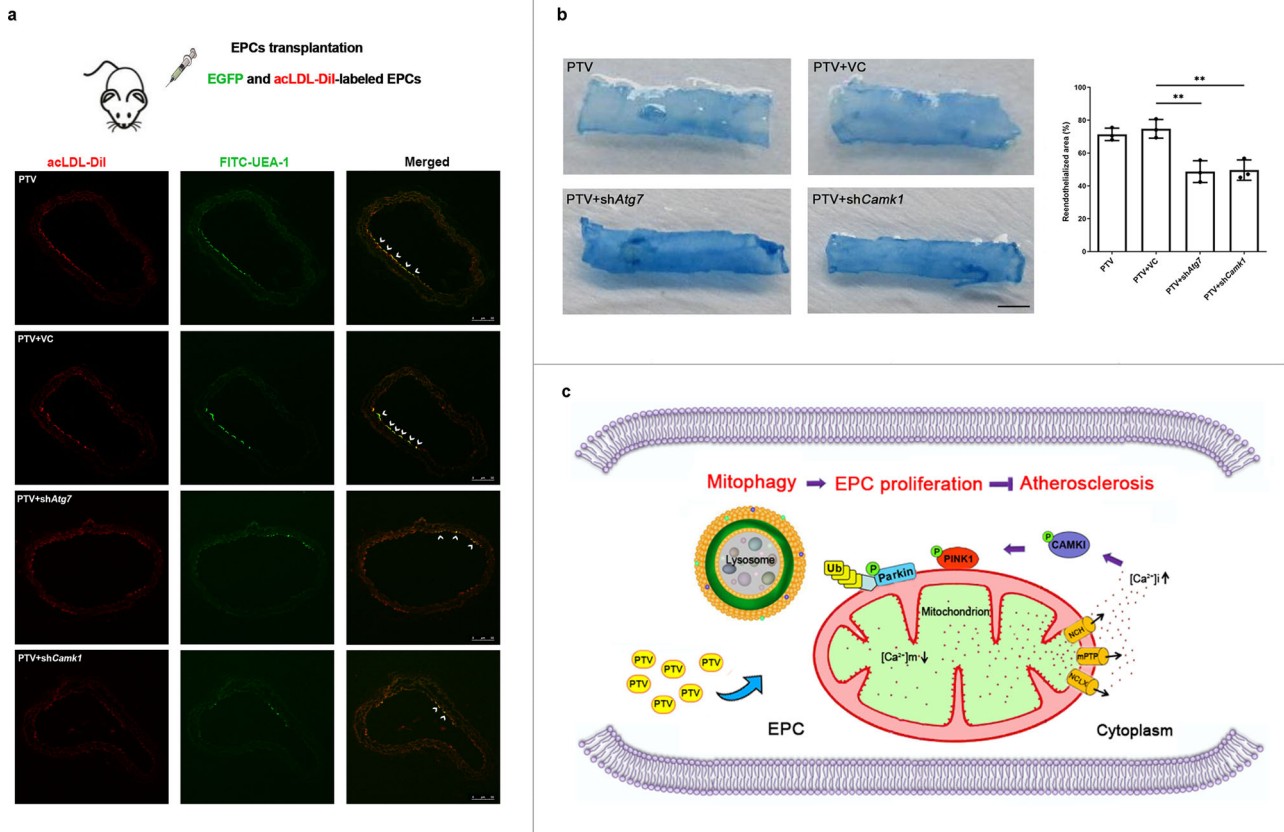

**Fig. 9 Disrupting of CAMK1-mediated mitophagy prevents EPC-mediated repairment of vascular endothelium. a** EPCs tracing in vivo. pEGFP-N2-EPCs were pre-treated by 0.5 µM PTV for 24 h and labeled by acLDL-DiI. EPCs in PTV + shAtg7 group were received lentivirus transfection to knock down Atg7, and EPCs in PTV + shAtg7 groups were received lentivirus transfection to knock down Camk1, respectively. Then EPCs were injected into *ApoE*$^{-/-}$ mice after vascular injury and attached to the vascular injury site. EPCs homed to vascular endothelium exhibited both red (acLDL-DiI) and green (EGFP) fluorescence. More EPCs in PTV + VC group home to vascular injury site compared with those in PTV + shAtg7 and PTV + shCAMK1 groups. **b** Quantification of Evans blue staining showed that knocked down Atg7 (49.63 ± 6.20 %) or Camk1 (48.70 ± 6.62%) reduced reendothelialization area compared with those in PTV + VC groups (74.73 ± 5.68 %). VC, vector control. (EPCs were isolated from 3 normal mice. The figures represented 3 independent experiments; mean ± SD, **$P$ < 0.01). **c** A schematic drawing of the calcium signal and mitophagy pathway in EPCs after PTV treatment. PTV elicits mitochondrial calcium release and increases [Ca$^{2+}$]$_i$ that further phosphorylates CAMK1. CAMK1 phosphorylation serves as a PINK1 kinase to activate PINK1 and recruit as well as phosphorylates PARK2 to induce mitophagy. Mitophagy activation protects EPC proliferation by reducing mitochondrial ROS production and maintaining mitochondrial homeostasis in atherosclerosis.

protein family on OMM, a group of mitophagy receptors that are transcriptionally regulated and engage in mitophagy receptor activity through the phosphorylation status of their LIR motif. This protein group contains BNIP3L/NIX[45–47] and FUNDC1[48]. In our study, we have screened the expression of both PINK1, PARK2, and BNIP3L. As PINK1-PARK2 rather than BNIP3L showed significant difference in atherosclerotic conditions as well as after PTV treatment, we infer that the PINK1-PARK2 pathway plays a major role in mitophagy induction in EPCs. As a mitochondrial serine/threonine kinase, PINK1 acts as a sensor and initiates the clearance of the damaged mitochondria under stress, particularly in Parkinson's disease[49] and ischemic acute kidney injury[50]. PINK1 has been reported to exert both autophosphorylation and phosphorylation activity, affecting cell damage by mitophagy regulation and signal transduction[51]. The role of phosphorylation at Ser[228] of PINK1 has been well-studied during MMP decrease, and this has been considered as the classical mitophagy activation pathway[32]. Notably, our research provided a finding that increased intracellular calcium concentration could activate PINK1. Together with the fact that both the phosphorylated form of PINK1 and PARK2 were decreased when sh*Camk1*, implying that the activation of CAMK1 may relate to

the phosphorylation of PINK1. Though in our study, we employed an in vitro kinase assay to exclude the possibilities of PINK1 phosphorylated by other proteins, there is still lack of sufficient evidence to support that CAMK1 could directly phosphorylate PINK1. Therefore, further study is needed to explore how CAMK1 activates PINK1 phosphorylation.

In the past few decades, CAMKs have been confirmed to play roles in synaptic plasticity, circuit development, and neuronal transmission. The best-characterized CAMKs are CAMK1, CAMK2 and CAMK4 through their neuronal functions[52]. CAMK1 is highly expressed in many human tissues and is central to the calmodulin-dependent protein kinase cascade and intracellular signal transduction such as endosome formation, Golgi to lysosome transport, ion transport, protein transmembrane transport as well as phosphorylation, and ubiquitin ligase activation[53]. Thus, we speculated that CAMK1 might also be associated with mitochondria to lysosome transport, despite little information regarding CAMK1 in autophagy or mitophagy regulation. This study reveals that CAMK1 contributed to PINK1 phosphorylation to induce mitophagy. Coupled with our recent research that shows that [Ca$^{2+}$]$_i$ accumulation results in targeting of CAMKK2 to activate autophagy in EPCs[15], this study further

illustrates signal transduction between the CAMK cascade and autophagy/mitophagy induction. Similar to our previous report, we also detected an increase of AMPK phosphorylation at Thr[172] site after PTV treatment (Supplementary Figure 3), in accordance with the reported vasculoprotective effects of statins through the AMPK activation pathway[54–56].

The role of statins as vasculoprotective agents is mostly attributed to their circulating cholesterol-lowering effect. However, over the decade, many clinical trials indicated statins can also act as anti-inflammatories, anti-oxidants, immunomodulators, and inhibitors of platelet aggregation[10–12]. This pleiotropy results from the inhibition of the synthesis of essential isoprenoid intermediates and inactivation of related signaling proteins. Statins were also reported to cause epigenetic modifications by promoting histone H3 and H4 acetylation or demethylation by DNMTs inhibition[57,58]. A recent study also found that simvastatin changed the expression of more than 400 miRNAs in different cell types. As individual miRNAs can potentially bind to several mRNAs, potential downstream effects are innumerable[59]. Our findings suggest that PTV reversed protective mitophagy through the PINK1-PARK2 pathway in atherosclerotic mice EPCs. Similarly, Allen et.al found that simvastatin-mediated cardioprotection was dependent on the upregulation of mitophagy through AKT-MTOR suppression in cardiomyocytes[11]. This effect was abolished by supplementation of mevalonate suggesting that it, or downstream isoprenoid intermediates, contributed to statin-induced autophagy or mitophagy in cardiomyocytes. Other reports have indicated that statins inhibit oxidative stress by CYBB, which enhances autophagy and improves muscle health and function in muscular dystrophy[60]. However, in this study, we demonstrate that PTV elicits calcium release and phosphorylates CAMK1, which further regulates the phosphorylation of PINK1 and PARK2 to induce mitophagy. PTV-induced mitophagy was successfully blocked by calcium chelators and Camk1 silencing. The different signaling pathways may be related to the variety of calcium channels and the calcium sensitivity between excited and non-excited cells.

Previous studies have demonstrated that statins improve EPC function in vivo and in vitro. Although not fully elucidated, the mechanisms may involve increased expression of NOS3[61] or annexin A2[9] and activation of the CXCL12-CXCR4[62] or phosphoinositide 3-kinase-AKT1[63] axes. We employed different concentrations of PTV to pretreat atherosclerotic mice EPCs and detected a consistently dose-dependent increase in proliferation, which peaked at 0.5 μM PTV (Fig. 3a). Due to this, 0.5 μM PTV for 24 h was chosen as the condition for following experiments. Recent reports suggested that statins might activate autophagy or mitophagy in different cell types. Hye Jin Wang et.al revealed that statins activated autophagic flux in hepatocytes, which was tightly coupled to hepatic gluconeogenesis and diabetogenic effects of stains[14]. Fogal V et al.[64] also reported that PTV suppresses glioblastoma cell proliferation through the induction of autophagy. We found that PTV not only improved atherosclerotic mice EPC proliferation but also reversed self-protective mitophagy. PTV-reversed mitophagy was closely related to atherosclerotic mice EPC proliferation recovery.

It is now well established that spatiotemporal changes of $[Ca^{2+}]_i$ are an important part of autophagy, but an exact mechanism remains elusive. Many reports consider calcium as an inhibitor of autophagy through ITPR3, that acts as a scaffold protein bound to BECN1 to form a complex. ITPR3-BECN1 formation decreases the level of free BECN1 in the cytoplasm and inhibits autophagy[26,27,65,66]. Conversely, others have shown that $[Ca^{2+}]_i$ increase by calcium ionophore ionomycin, ATP, vitamin D, and $Cd^{2+}$ activated autophagy flux, which was successfully blocked by calcium cheltors[24,25]. This effect might be related to

the activation of CAMKs, AMPK and ERK protein kinases after $[Ca^{2+}]_i$ increase. Last year, we confirmed that SOCE-mediated calcium influx activated EPC autophagy through the CAMKK2-MTOR pathway[15]. In this study, we further reveal that mitochondrial calcium release also upregulates mitophagy, but mitochondrial $Ca^{2+}$ extrusion mechanisms have not been completely elucidated. The mitochondrial sodium-calcium exchanger (NCLX) has been considered to have a major role in excitable cells, but a recent study confirmed that NCLX balances calcium homeostatsis in ECs too[67]. In addition, simvastatin triggers a weak mitochondrial $Ca^{2+}$ efflux through NCLX in skeletal muscle[28]. Another possible mechanism depends on the ubiquitous hydrion-calcium exchanger (HCX), the identity of which is still controversial[68]. In addition, the mitochondrial permeability transition pore (mPTP) has non-specific channel property that also contributes to mitochondrial calcium extrusion, but its molecular composition is not yet clear[69]. EPCs work as non-excited cells, it is likely that NCLX is responsible for PTV-elicited mitochondrial calcium release in EPCs. Of course, this needs to be clarified in future studies.

In conclusion, as shown in the schematic drawing (Fig. 9c), EPC proliferation and mitophagy are inhibited in atherosclerotic mice. PTV promotes mitochondrial calcium release and reverses EPC mitophagy, contributing to an improvement in proliferation. The activated mitophagy relied on the mitochondrial calcium release activated by CAMK1, as well as CAMK1-dependent phosphorylation of PINK1 and PARK2.

Our results represent evidence for mechanism for the effect of statins on EPCs through protective mitophagy activation. Enhancing protective mitophagy improves EPC survival rate, protects proliferation, and promotes vascular re-endothelialization to slow down the progress of atherosclerosis. Furthermore, modification of $[Ca^{2+}]_i$ through mitochondrial calcium release and the activation of CAMK1 may contribute to a therapeutic breakthrough promoting protective mitophagy regulation. Our findings may provide a proliferation-promoting and mitophagy-activating mechanism as well as a therapeutic target in the EPCs of atherosclerotic patients.

## Methods

**Antibodies and media.** Antibodies against mouse MAP1LC3B (2775S), SQSTM1 (5114S), PINK1 (6946S), MFN2 (9482S), ATG7 (8558S), BNIP3L/NIX (12396) were purchased from Cell Signal Technology. Antibody for PINK1 (ab23707), PARK2 (ab77924), TOMM20 (ab186734), phospho-Thr[177]-CAMK1 (ab62215), CAMK1 (ab68234), AMPK (ab32047) and phospho-Thr[172]-AMPK (ab133448) were obtained from Abcam. Antibody for phospho-Ser[65]-PARK2 (bs-19882R) was purchased from Bioss. Antibody for phosphor-Ser[228]-PINK1 (AF7081) was obtained from Affinity Biosciences. Antibody for ATG5 (AP1812a) was from Abgent and the antibody could detect both ATG12-ATG5 conjugate (molecular mass was 55 kDa) as well as free ATG5 (molecular mass was 32 kDa). Antibodies for flow cytometry test were as follows: APC anti-mouse CD64 (139305, Biolegend), BV421 anti-mouse CD16/32 (101332, Biolegend), PE anti-mouse c-Kit (12-1171-82, ebioscience), PE-Cy7 anti-mouse CD41 (25-0411-80, ebioscience), FITC anti-mouse CD127 (11-1271-81, ebioscience), FITC anti-mouse CD93 (11-5892-81, ebioscience).

EGM-2MV BulletKit medium from Lonza included endothelial basal medium (EBM-2, CC-3156) and 10% fetal bovine serum (CC-4101A), recombinant Homo sapiens (rHs) IGF1 (CC-4115A), rHsVEGF (CC-4414A), rHsEGF (CC-4317A), rHsFGF2/FGF-B (CC-4101A), heparin (CC-4396A) and ascorbic acid (CC-4116A. Lymphoprep (1.083, 10831), ionomycin (407950), bafilomycin A1 (BAFA1, B1793), FITC-UEA-I (L9006) and 3-methyladenine (3-MA, M9281) were purchased from Sigma. Pitavastatin (PTV, B1124) was obtained from ApexBio Technology.

**Isolation and characterization of EPCs.** All animal procedures were purchased from the Animal Center of Third Military Medical University (Army Medical University) and approved by the Experimental Animal Ethics Committee of the Third Military Medical University before performing the study and conformed to the regulations of Guide for the Care and Use of Laboratory Animals (8th edition, National Research Council, USA, 2011). Isolation and characterization of EPCs were performed as previously reported[15]. In brief, male $ApoE^{-/-}$ mice (8 or

16 weeks age) were anesthetized with an intramuscular injection of 5 mg/kg xylazine and 100 mg/kg ketamine, then sacrificed by cervical dislocation. Bone marrow was harvested from the tibias as well as femurs of mice. Bone marrow-derived mononuclear cells (BMNCs) were isolated by density-gradient centrifugation. At last, the BMNCs were cultivated in EGM-2MV BulletKit medium. As described in our recent works[15], briefly, characteristic phenotypes of EPCs were identified by immunostained with DiI-acLDL and UEA-I lectin. The treble-positive cells were then analyzed for the expression of CD133, CD34, VEGFR2, and CD31 by flow cytometry.

**Blood routine test**. To assess circulating immune cell content in atherosclerotic mice, peripheral blood was obtained from the orbital sinuses, and 20 μl of whole blood was mixed with 200 μl diluent immediately. The blood cells of the mice were analyzed using Mindray BC-5300 Vet animal automatic hematology analyzer.

**Bone marrow cells isolation and flow cytometric analysis**. Bone marrow was harvested by flushing the femurs and filtered through 40 μm filters. BMNCs were isolated using density-gradient centrifugation followed by washing 3 times in PBS. Next, collected cells were stained with the mixed antibodies for flow cytometry. After staining with the listed markers for 45 min in the dark at 4 °C, samples were washed twice, and transferred to a cell strainer to obtain a single-cell suspension. Finally, cells were analyzed using a flow cytometer (Beckman Gallios).

**Immunohistochemistry**. Aortic root was snap-froze in optimal cutting temperature compound, and 4 um frozen sections were cut and the endogenous peroxidase activity blocked with Hematoxylin staining solution (P1000A, Beyotime Biotechnology) for 10 min. Antigen retrieval was performed with quick antigen retrieval solution (P0090, Beyotime Biotechnology), then sections were blocked with 5% normal serum, and incubated with the primary antibody (1:100, ab125212, Abcam) for 24 h at 4 °C, followed by biotinylated secondary antibody (P0101, Beyotime Biotechnology) diluted 1:100, and counterstain with hematoxylin (C0107, Beyotime Biotechnology) for 1 min. Clear the tissue slides in 3 times of xylene and coverslip using mounting solution, then sections were observed under microscopy.

**Cell proliferation assays**. Cell proliferation was checked by the xCelligence Real-Time Cell Analyzer instrument (RTCA, ACEA Biosciences, San Diego, CA, USA). Measurement of cell proliferation was described in detail in our previous study[15]. In addition, we applied traditional cell counting kit-8 (CCK8, Beyotime biotechnology, C0038) to evaluate EPC proliferation. EPCs were plated on the 96-well culture plate and underwent different treatments, before addition of WST-8 dye (10 μl). Cells were incubated at 37 °C for 4 h and absorbance at 450 nm was measured with a microplate reader.

**Fluorescence $Ca^{2+}$ measurements in cytoplasm and mitochondria in intact cells**. We employed fluo3-AM (S1056, Beyotime Biotechnology) to detect the concentration of cytoplasm in intact cells. The cells were cultured in a glass-bottomed dish for 5 days and then washed with $Ca^{2+}$ free HBSS 3 times, before incubation in $Ca^{2+}$ free HBSS containing 5 nM fluo3-AM at 37 °C for 40 min in the dark. Cells were washed twice and green fluorescence observed by laser scanning confocal microscope (LSCM). To measure mitochondria, cells were loaded with 15 μM Rhod2-AM (R1234MP, Molecular probes) at 37 °C for 1 h in the dark, before being washed with $Ca^{2+}$ free HBSS twice, and imaged under LSCM for 1 h. Fluorescent intensity F was normalized to the baseline fluorescence value $F_0$ ($F/F_0$) and expressed as $[Ca^{2+}]_i$. To quantify $[Ca^{2+}]_i$ in EPCs, we measured $F_{max}$ and $F_{min}$ of $[Ca^{2+}]_i$ as previously described[15]. $F_{max}$ was obtained by perfusion with 10 μM ionomycin and 5 mM $CaCl_2$; $F_{min}$ was measured by perfusion with 10 mM EGTA and 20 μM BAPTA-AM (B1205, Molecular probes) in HBSS. $[Ca^{2+}]_i$ was derived after in situ calibration according to the following equation.

$$[Ca^{2+}]_i(nM) = K_d \times (F - F_{min})/(F_{max} - F). \quad (1)$$

The $K_d$ of fluo3 for $Ca^{2+}$ at room temperature is 400 nM, while the $K_d$ of Rhod2-AM at room temperature is 570 nM.

**GFP-mRFP-LC3 adenoviral vector monitor the autophagy flux**. EPCs were infected with an adenoviral vector containing GFP-mRFP-LC3 (HanBio Technology) according to the manufacturer's instruction. After infection for 12 h, medium was replaced and cells were incubated for 24 h. We evaluated infection efficiency and autophagy flux under LSCM by counting the number of RFP and YFP puncta.

**Measurement of mitophagy levels using the mtKeima reporter**. Mitochondria-targeted monomeric Keima-Red (mtKeima) assay (Medical and Biological laboratories Co., Ltd. AM-V0251HM) was used to evaluate mitophagy levels in EPCs. EPCs were transfected with mtKeima plasmid and stable cell lines were selected by hygromycin. For the assay, mtKeima expressing cells were cultured on a glass-bottomed dish for the next experiments. EPCs were imaged on LSCM (Leica TCS-SP5). Excitation wavelengths and emission filters used were as follow:

Cytoplasmic Keima: 488 nm, 650–760 nm; Lysosomal Keima: 561 nm, 570–630 nm; Hoechst: 375 nm, 435–480 nm. High ratio regions were automatically segmented, and their areas were calculated. The whole-cell region was delineated manually on a fluorescent image to facilitate area calculation. The proportion of the high ratio (561 nm/488 nm) signal area (red) to the total mitochondrial area was served as the mitophagy index.

**Gene silencing**. EPCs were cultured on a glass-bottomed dish for 5 days, before addition of lentivirus vector (LV) carrying different shRNA to the medium at a multiplicity of infection of 100. LVs contained shRNA targeted to Atg7 (Hanbio Technology, Shanghai, China), Camk1, Pink1, or Park2 (Gene Pharma, Shanghai, China). After 48 h, transfection media was replaced with fresh media and incubated for 24 h. Western blots were used to measure the efficiency of silencing Atg7, Camk1, Pink1, and Park2 in EPCs.

**Pink1 knockout mouse model**. The Pink1 knockout (KO) mouse model (C57BL/6J) was created by CRISPR/Cas-mediated genome engineering (Cyagen Biosciences). Cas9 and gRNA were co-injected into fertilized eggs and thus disrupting the main kinase domain due to the reading frame shift. The established 8-week Pink1 KO mice have exons 4~7 deleted as target site, which covers 40.92% (~2582 bp) of the coding region.

**Mitochondrial membrane potential analysis**. Mitochondrial membrane potential (MMP) was measured by MMP assay kit and JC-1, a marker of mitochondrial activity (C2006, Beyotime Biotechnology). JC-1 accumulates and aggregates in polarized mitochondria (red), and becomes monomeric (green) and retained in cytosol when MMP is lost, so a decrease of red/green fluorescence intensity ratio represents depolarization. Briefly, EPCs were collected and incubated with 0.5 ml JC-1 working solution for 25 min at 37 °C (in the dark) and then washed in cold JC-1 staining buffer twice. Cells were then resuspended in medium. Red fluorescence intensity was measured at 525 nm (ex) and 590 nm (em) and green fluorescence excitation at 490 nm (ex) and 530 nm (em) using LSCM.

**Mitochondrial ROS measurement**. Mitochondrial ROS was measured with MitoSOX red (M36008, Molecular Probes, Eugene, OR, USA). Briefly, cells in each group were incubated for 15 min at 37 °C in 0.5 ml measurement buffer containing 5 mM MitoSOX Red. Cells were then washed twice with PBS and observed under LSCM.

**Mitochondrial morphology by transmission electron microscopy**. EPCs were collected, fixed in 2.5% glutaraldehyde at 4 °C for 2 h, and immersed in 1% osmium tetroxide for 2 h. Fixed cells were washed in PBS, dehydrated in acetone, and embedded in Epon 812 (SPI Supplies, West Chester, PA, USA). After slicing the samples were observed under a JEM-1400PLUS TEM (JEOL, Tokyo, Japan) operating at 100 kV.

**Immunoblotting**. EPCs were harvested and rinsed 3 times in ice-cold PBS. Then, cells were lysed by cell lysis buffer (89900, Pierce) containing 2 mM sodium orthovanadate and 0.5 mM PMSF. EPCs were centrifugated in 14000 g for 15 min and protein concentration measured using the BCA assay (P0012, Beyotime Biotechnology). Total protein was separated by SDS-PAGE and transferred to PVDF membranes. The membrane was blocked at 37 °C for 1 h with 5% non-fat milk and incubated with primary antibody diluted 1:100 overnight at 4 °C. Then, we washed the membrane 3 times with TBS (AR0031, Boster Biological Technology) with 0.5% Tween-20 (T8220, Solarbio). The membranes were continuously incubated with HRP-conjugated secondary antibodies at 37 °C for 2 h. Bands of protein were visualized by chemiluminescence detection and quantified by Image Quant TL software (GE Healthcare, Sweden).

**Immunofluorescence**. EPCs were stained with MitoTracker Deep Red FM (500 nM, Invitrogen) and fixed in 4% paraformaldehyde (Beyotime Institute of Biotechnology) at room temperature for 10 min. After two PBS washes, EPCs were permeabilized by 0.1% Triton 100-X (Beyotime Institute of Biotechnology) at room temperature for 15 min. Cells were washed with PBS three times and blocked in Blocking Buffer for Immuno-Staining (Beyotime Institute of Biotechnology) at 37 °C for 30 min. Samples were then incubated overnight at 4 °C with anti-MAP1LC3B antibody (1:50) or anti-PARK2 antibody (1:50) and then washed in PBS twice, before staining with secondary antibody (1:500) at 37 °C for 2 h. Colocalization of PARK2-mitochondria and MAP1LC3-mitochondria was measured at 100-400 Hz under the LSCM (40 ×1.25 oil objective, Leica TCS-SP5). All images obtained from LSCM have adjusted background signal using the samples without primary antibodies as the negative controls. Images were analyzed by LAS X software (Leica) and Image-Pro Plus 5.0 (Media Cybernetics) and represented as a Pearson's coefficient.

**In vitro kinase assay**. An in vitro kinase assay was performed as previous published[70]. In brief, 1μ g recombinant PINK1 protein (denatured) (ab116177) was

incubated in the presence or absence of 25 ng CAMK1 for 10 min at 30 °C with the following additions: 10 mM $MgCl_2$, 0.2 mM ATP, 1 mM $CaCl_2$, and 1 μM CaM in 50 μl reaction system. Reactions were terminated by boiling in SDS-2-ME dissociation solution and analyzed by immunoblot.

**EPC transplantation and tracing in vivo.** To observe whether the pre-treated EPCs were capable of homing to the site of injury and showing better proliferation capacity, enhanced green fluorescent protein (EGFP)-labeled EPCs were marked by acLDL-DiI (Invitrogen, CA, USA) for 1 h. Then, 200 μl EGFP and acLDL-DiI-labeled EPCs ($1 \times 10^6$) were injected to mice via tail vein. 7 days later, EPC tracking and immunohistochemistry were performed. Images of the stained cells were obtained by a fluorescence microscope (Leica TCS-SP5).

**Measurement of reendothelialization.** Endothelial regeneration was evaluated by staining the denuded areas by injecting 200 μl of 5 % Evans Blue dye with saline via the tail vein into the heart. The left common carotid artery was then harvested 5 mm away from the carotid bifurcation. The reendothelialized area appeared white in color (unstained), whereas the non-endothelialized lesions appeared blue (stained). The unstained areas (in white) and the total carotid artery areas were measured. The ratio of reendothelialized areas (unstained area) versus the total carotid artery area was calculated.

**Statistics and reproducibility.** All analyses were performed using SPSS 19.0 software. The measurement variables are presented as the mean ± standard deviation (SD). Significance was determined using t-test corrected for multiple comparisons (Least-Significant Difference). Nonparametric ANOVA (Kruskal–Wallis) followed by the Dunn multiple comparison post-hoc test was used when one or more datasets showed non-normal distribution. Number of biological replicates and observations were described in the figure legends. Statistical significance was considered at $P < 0.05$, with $*P < 0.05$; $**P < 0.01$. For graphs, all data were analyzed using GraphPad Prism software (version 5.0 or 8.4.0).

**Reporting Summary.** Further information on research design is available in the Nature Research Reporting Summary linked to this article.

## Data availability
All data support the main and supplementary figures are either available online, or available from the corresponding authors upon reasonable request. Source data behind the graphs can be found in Supplementary Data 1.

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

## Acknowledgements

This work was supported by the National Natural Science Foundation of China under grants (81800396, 82170323). Chongqing Talent Project (Yang Jie) and Chongqing Natural Science Foundation (cstc2019jcyj-msxmX0471). Innovation Project of Medical Subject (310901SZXK).

## Author contributions

L.H. and J.Y. conceived the approach. L.H. provided the overall supervision of the project. The development of methodology was performed by J.Y., M.J.S., and H.T. X.B.G. contributed to the validation of the overall replication. Y.Q.Y. and R.C. specifically performed the experiments. Statistical analysis and generation of figures were carried out by R.Z.C., C.L., and J.H.Z. All authors interpreted the results. J.Y., M.J.S., and H.T. wrote the first manuscript. L.H. and J.Y. wrote the final manuscript on which all authors commented and RC contributed to the process of manuscript revision.

## Competing interests

The authors declare no competing interests.
