## [Transparent Peer Review File · Communications Biology]

Reviewers' comments:

Reviewer #1 (Remarks to the Author):

The current study aims to establish that Pitavastatin activates mitophagy to protect endothelial progenitor cell proliferation through a calcium-dependent CAMK1-PINK1 pathway in atherosclerotic mice.

While perhaps interesting observations are presented in terms of autophagy and calcium, in particular the mitophagy aspects of the study seem less well developed. This is in part due to the choice of model systems and tools that at least at this point do not unequivocally establish a role for PINK1-PARK2 mitophagy. Also there are quite a few other claims that are not necessarily substantiated through the presented data and are rather misleading. For instance that CAMK1 served as a PINK1 kinase suggests it would phosphorylate PINK1, which isn't really shown.

Specific comments:

There is a notorious lack of reliable Pink1 and Park2 antibodies, especially for mouse. With the exception of PRK8 (at least for western blots), most of the antibodies have not been widely used in the field yet and as such need to be established first. The authors should perform additional experiments to demonstrate their specificities.

On the conceptual level, one would not necessarily expect to see much Pink1 protein under basal conditions (normal diet). Also due to their particular relation, typically Park2 protein levels go up when Pink1 levels are down and vice versa. With that said, and given the reduced mitochondrial membrane potential and increased damage one would expect to see more Pink1 accumulation, not less. Downstream there could be less autophagic flux and thus overall less mitophagy.

An intensity-based mtKeima readout might be more appreciate than 'per area' especially as the fluorescence signal does not look particularly mitochondrial at least not on the images provided. Higher resolution images are certainly required for the mtKeima data.

In contrast to shRNA knockdown, Pink1 or Park2 knockout animals could really provide a much clearer answer than the knockdown approach. While there seems to be reduced signal for both, Pink1 and Park2, the remaining levels should still be sufficient to execute mitophagy as mostly complete loss of either is needed for noticeable changes.

Other general concerns:

MMP and ROS analyses upon PTV treatment and/or shRNA manipulations not missing.

All western blots critically need molecular weight markers and ideally the respective full-length (uncropped) blots would be included on the supplement as thus could provide more confidence.

There are some inconsistencies noted between western blots and quantifications or data across the figures. For instance, Figure 5B and C/D. Another example is Figure 8 where levels of total Pink1 or Park2 do not change (in contrast to Figure 5) but only the phosphorylated forms increase with PTV treatment.

Other essential controls seem missing such as no primary antibodies for staining in order to exclude any potential signal bleeding through the channels.

Reviewer #2 (Remarks to the Author):

The submitted manuscript titled "Pitavastatin activates mitophagy to protect EPC proliferation through a calcium-dependent CAMK1-PINK1 pathway in atherosclerotic mice" is a scientifically sound research article that aims to investigate the mechanism of pitavastatin-activated mitophagy in endothelial progenitor cell proliferation. The findings reported and further developed in its very well-written discussion provide a certain insight into the role of mitochondria and mitophagy in atherogenesis and are of considerable scientific interest for this specific research field, while also falling within the scope of the journal. This is a clear and interesting article analyzing calcium-dependent pathways in atherosclerotic mice. Strikingly, the manuscript also highlights novel approach to anti-atherosclerotic therapy based on mitophagy activation. The paper is well-written and documented, and contains an interesting discussion.

Reviewer #3 (Remarks to the Author):

This study constitutes novel work exploring how an EPC proliferation is improved in pitavastatin treatment to improve the endothelial mitophagy dysfunction against atherosclerosis. However several recent reports has evaluated the effect of pitavastatin on atherosclerosis and endothelium (namely Changqin Jing et al., 2017 and Yoichi Morofuji et al., 2010). Therefore, although it is always useful for independent groups to reassess the nuances of pitavastatin on such vascular diseases, this paper unfortunately conducts only cursory evaluation of these pathologies and leaves many questions unanswered/unevaluated.

Reviewers' comments:

Reviewer #1 (Remarks to the Author):

The authors provide a somewhat improved manuscript, but the main concerns unfortunately remained rather unaddressed. Aside from potential misleading claims, the specificity of some reagents and measurements must be demonstrated and there is need to substantiate a specific role of Pink1-Park2 mitophagy over more general effects on mitochondria and global autophagy.

- At a minimum, authors should rephrase their statement on CAMK1-dependent phosphorylation of PINK1 and PARK2. It must be made clear that CAMK1 is not directly phosphorylating these proteins, at least this is not shown here.
- The specificities of the antibody tools especially for mouse cells/tissue remains to be established. The Pink1 KO mouse would be the best model to demonstrate the validity of the (phospho-) Pink1 and Park2 antibodies. The model however is only being used for EPC proliferation assessment, which is a missed opportunity to solidify the various mitophagy aspects.
- Also while most westerns are now shown uncropped, this is not the case for the shRNA knockdown of Pin1k, Park2, and Camk1 that are the most critical of all.
- It remains uncertain why the authors chose to employ a mitophagy index that seems to be a readout of total area occupied by mitochondria, as opposed to a fluorescence intensity measure that is typically being used for mitoKeima.

Reviewer #4 (Remarks to the Author):

Overall this is a good study devoted to the research of pitavastatin effect on endothelial progenitor cells (EPCs). The authors found convincing evidences that pitavastatin is capable to induce mitophagy in EPCs from mice using different approaches to prove it. It should be noted that there are more and more evidences about the role of mitochondrial mutations in atherosclerosis development (and in the development of many other diseases ranging from cancer to pulmonary pathologies). Thus, the discoveries made in this paper about effects of pitavastatin on mitophagy deserves to be shown to scientific community. The authors improved the manuscript as it could be judged by reading the replies to the reviewer's comments and marked sections of the manuscript. The experiments conducted were adequate to tasks and the results obtained are organized in a logical manner. The introduction is enough to understand the tasks. The discussion section properly analyses the data obtained and provides conclusions.

While I do not criticize most of the experiments done by the authors I will have to mention certain issues attracted my attention.

The authors do not clarify or justify why they chose pitavastatin among other statins. Also there is no mention about origin or the manufacturer of pitavastatin used.

In the proposed mechanism of pitavastatin-induced mitophagy (Fig. 10) the initial process starting mitophagy is mitochondrial calcium release. The authors should provide some ideas about the potential mechanism by which pitavastatin causes calcium release. Otherwise the mechanism of pitavastatin-related mitophagy induction will not be fully understood (at least, hypothetical mechanism).

The authors also should mention in the manuscript the role of mitochondrial DNA mutations in atherosclerosis and provide some references. The mitophagy seems to be an important process to help cells to get rid from impaired mitochondria with mutations.

"After 7 days of transplantation, labeled cells were traced to home and showed red (acLDL-DiI) and green fluorescence in line in carotid artery intima (Fig. 9A, white arrow)."

There is no white arrow in Fig.9A.

English language should be corrected in some cases:

"The use of statins as vasculoprotective agents are mostly due to reductions in circulating cholesterol."

"PTV was also reported to inhibited glioblastoma cell proliferation through autophagy induction⁵⁸."

Overall it is a well written paper with great results and I recommend it for publication after minor revision related to above mentioned questions/suggestions.

Reviewer #3, previous round of review:

In this manuscript, the authors link the role of pitavastatin to the activation of mitophagy to protect EPC proliferation in atherosclerosis model. First they show in the mice with atherogenesis that EPC proliferation decreased and directly correlated with mitochondrial dysfunction suggestive of mitophagy impairment. Using PTV they show reversions in mitophagy function and they show CAMK1 induce with PINK1-PARK2 kinase to induce mitophagy in the EPC from atherosclerotic mice. Finally, in the carotid artery intima injury model, they show pitavastatin can prevent EPC-mediated repair of vascular endothelium.

Overall comments: This study constitutes novel work exploring how an EPC proliferation is improved in pitavastatin treatment to improve the endothelial mitophagy dysfunction against atherosclerosis. However several recent reports have evaluated the effect of pitavastatin on atherosclerosis and endothelium (namely Changqin Jing et al., 2017 and Yoichi Morofuji et al., 2010). Therefore, although it is always useful for independent groups to reassess the nuances of pitavastatin on such vascular diseases, this paper unfortunately conducts only cursory evaluation of these pathologies and leaves many questions unanswered/unevaluated.

Specific suggestions to improve this manuscript are as follows:

Major points:

1) Atherosclerogenesis is largely an assessment of endothelial cell proliferation with immune cell accumulation in early stage. Also VSMC proliferation and migration into the intima appears in late stage. It appears that the authors completely neglected this fact. Clearly, their model is begging for further followup experiments at the vessel wall.

a) On histology of their atherosclerotic model or ex vivo EPC culture, is there no immune cells as would be expected?

b) Is there no immune cells proliferation as would be expected?

c) How could you ignore the other cell type in this your endothelial cell progenitor?

d) If they are thinking that EPC proliferation is affected in the atherogenic lesions, they can do the RNA sequencing of these vessels for these markers.

2) Authors use only LC3 as the autophagy markers of choice. I am not sure why they only use LC3 as the autophagy marker. Have you checked the other autophagy marker such as p62? Since LC3 goes up when pitavastatin treatment, the question arises whether all autophagy markers are going up or "flux". If most markers are going up, then this is less consistent with flux but rather it is a transcriptional response by TFEB, TFE3, or other master regulators. They can ascertain this by doing qPCR of the same markers to see if they go up. If the transcripts go up, then the authors are really not assessing "flux". For "flux", the authors will need to use agents such as Bafilomycin or chloroquine to see if LC3 and p62 change accordingly.

3) However, the big issue that arises here is again, why is pitavastatin, which affects mitophagy, having any effect on global autophagy? This piece of the paper needs to be explained and unraveled.

4) How is atherosclerosis reduced when pitavastatin is given? Any changes to atherosclerosis in

pivastain administration is likely secondary effects unless the authors can show pivastain actually tangibly affects tissue levels.

Dear Editor,

Further to our correspondence a couple of months ago, we are attaching the revised version of the article entitled “Pitavastatin activates mitophagy to protect EPC proliferation through a calcium-dependent CAMK1-PINK1 pathway in atherosclerotic mice” again. We have completed all of the changes according to your and reviewers’ comments. To make the changes easier to identify where necessary, we have numbered them and responded specifically to each suggestion below. Changes to the manuscript are shown in red to denote where we made revisions.

Reviewer #1 (Remarks to the Author):

1. At a minimum, authors should rephrase their statement on CAMK1-dependent phosphorylation of PINK1 and PARK2. It must be made clear that CAMK1 is not directly phosphorylating these proteins, at least this is not shown here.

Responses: According to your suggestion, we used an in vitro kinase assay to support that CAMKI regulates the PINK1-PARK2 pathway by functioning as a PINK1 kinase. Indeed, in vitro kinase assay demonstrated that CAMK1 is a key serine kinase activated by elevated Ca^{2+} and catalyzes phosphorylation of PINK1 on serine²²⁸ directly. The corresponding data were shown in **supplementary fig. 5A** and we have added this in Results section.

2. The specificities of the antibody tools especially for mouse cells/tissue remains to be established. The Pink1 KO mouse would be the best model to demonstrate the validity of the (phospho-) Pink1 and Park2 antibodies. The model however is only being used for EPC proliferation assessment, which is a missed opportunity to solidify the various mitophagy aspects.

Responses: Thanks for your helpful suggestion. We retested the critical protein (PINK1 and PARK2) from *Pink1* KO mice to assess the antibodies (**Fig. 6A**). Besides EPC proliferation, we evaluated mitophagy in *Pink1* KO mice (**Fig. 6B-6D**). The relevant results were shown in the revised manuscript.

3. Also while most westerns are now shown uncropped, this is not the case for the shRNA knockdown of Pin1k, Park2, and Camk1 that are the most critical of all.

Responses: According to your advice, we have provided the uncropped blots of knock down *Pink1*, *Park2*, and *Camk1*. New images were replaced in **Fig.5A** (*shPink1* and *shPark2*) and **Fig. 8A** (*shCamk1*).

4. It remains uncertain why the authors chose to employ a mitophagy index that seems to be a readout of total area occupied by mitochondria, as opposed to a fluorescence intensity measure that is typically being used for mitoKeima.

Responses: Thanks for your comment. As pervious literatures described ^[1], mitophagy index was calculated by analyzing the ratio (561nm/488nm) images of Keima. The area, but not fluorescence intensity was segmented and calculated respectively. We have revised a detailed description in the methodology for better distinguished it from the regular immunostaining analysis.

Reviewer #4 (Remarks to the Author):

1. The authors do not clarify or justify why they chose pitavastatin among other statins. Also there is no mention about origin or the manufacturer of pitavastatin used.

Responses: Thanks for your kind reminding. Pitavastatin as a member of the statin drug family is widely used for the treatment of hypercholesterolem. It shares similar biological pleiotropy like other commonly-used statins (such as atorvastatin and simvastatin) in pervious cohort study ^[2]. However, the exact mechanisms above are not well illustrated. Therefore, we choose pitavastatin to explore how it benefits the EPC proliferation in atherosclerosis mice model. Pitavastatin (PTV, B1124) was obtained from ApexBio Technology and we added this point in methodology.

2. In the proposed mechanism of pitavastatin-induced mitophagy (Fig. 10) the initial process starting mitophagy is mitochondrial calcium release. The authors should provide some ideas about the potential mechanism by which pitavastatin causes calcium release. Otherwise the mechanism of pitavastatin-related mitophagy induction will not be fully understood (at least,

hypothetical mechanism).

Responses: Thanks for your helpful advice. According to your suggestion, in this work, we have provided potential mechanism that pitavastatin-induced mitochondrial calcium release in the Discussion section. Firstly, mitochondrial sodium-calcium exchanger (NCLX) has been considered to have a major role in excitable cells, but a recent study confirmed that NCLX balances calcium homeostasis in endothelial cells too^[3]. In addition, simvastatin triggers a weak mitochondrial Ca²⁺ efflux through NCLX in skeletal muscle^[4]. Another possible mechanism depends on the ubiquitous hydron-calcium exchanger (HCX), the identity of which is still controversial^[5]. Moreover, the mitochondrial permeability transition pore (mPTP) has non-specific channel property that also contributes to mitochondrial calcium extrusion, but its molecular composition is not yet clear^[6]. We speculated that EPCs work as non-excited cells, it is likely that NCLX is responsible for PTV-elicited mitochondrial calcium release in EPCs. Of course, this needs to be clarified in future studies. We have added this point in the discussion section.

3. The authors also should mention in the manuscript the role of mitochondrial DNA mutations in atherosclerosis and provide some references. The mitophagy seems to be an important process to help cells to get rid from impaired mitochondria with mutations.

Responses: There is a growing body of evidence support that mitochondrial DNA (mtDNA) damage is related to the pathogenesis of atherosclerosis. In the present study, PTV-induced mitophagy get rid of the accumulated oxidative damaged mitochondria, which may partially contain the mitochondria with mutations. We have included this point in the revised manuscript.

**4. "After 7 days of transplantation, labeled cells were traced to home and showed red (acLDL-DiI) and green fluorescence in line in carotid artery intima (Fig. 9A, white arrow)."
There is no white arrow in Fig.9A.**

Responses: Thanks for your careful checks. We have added the white arrows (now **Fig. 10A**).

5. English language should be corrected in some cases:

"The use of statins as vasculoprotective agents are mostly due to reductions in circulating cholesterol."

"PTV was also reported to inhibited glioblastoma cell proliferation through autophagy induction⁵⁸."

Responses: Thanks for your correction. And we have corrected it as follows in the revised manuscript.

"The role of statins as vasculoprotective agents are mostly attributed to their circulating cholesterol -lowering effect. "

" Fogal V et al. also reported that PTV suppresses glioblastoma cell proliferation through the induction of autophagy."

Reviewer #3, previous round of review:

1. Atherosclerosis is largely an assessment of endothelial cell proliferation with immune cell accumulation in early stage. Also VSMC proliferation and migration into the intima appears in late stage. It appears that the authors completely neglected this fact. Clearly, their model is begging for further follow up experiments at the vessel wall.

a) On histology of their atherosclerotic model or ex vivo EPC culture, is there no immune cells as would be expected?

Responses: We have focused on the immune cells state in circulating and bone marrow of the established atherosclerotic mice. Previous studies have widely revealed the accumulation and interplay of pro- and anti-inflammatory immune cells in the arterial during plaque progression. Consistent with which, we as well verified more CD68 positive (stained for macrophages) in the lesion of atherosclerosis mice (**Supplementary fig.2**).

b) Is there no immune cells proliferation as would be expected?

Responses: The present work focused on how EPCs play the vasculoprotective role in atherosclerosis, rather than the proliferative and apoptotic dynamics of these immune cells. So, we have not further examined the proliferation markers of immune cells in the present study.

c) How could you ignore the other cell type in this your endothelial cell progenitor?

Responses: Thank you for the kind advice. We combined the co-immunofluorescence and flow cytometry to accurately identify and guaranteed the purity of EPCs we isolated, which was supplied in the method part.

Figure : Characteristic phenotypes of EPCs. (A) BMNCs plated on glass-bottomed cell culture dishes were immunostained with DiI-acLDL (red) and FITC-labeled UEA-I lectin (green). Cell nuclei were stained with DAPI (blue). The treble-positive cells were considered as EPCs, which accounted for 92% of total BMNCs (n = 98 cells). Scale bar: 250 μm (B) Adherent BMNCs were analyzed for the expression of PROM1/CD133, CD34, KDR/VEGFR2 and PECAM1/CD31 by flow cytometry. Dotted histograms represent isotype controls. (Cells were isolated from 3 mice for 1 experiment and figures were representative of 3 independent experiments.)

d) If they are thinking that EPC proliferation is affected in the atherogenic lesions, they can do the RNA sequencing of these vessels for these markers.

Responses: Thank you for mentioning a possible method to whether EPC proliferation could be identified as a possible mode of the atherosclerotic progression. We plan to characterized phenotypic and transcriptional diversity of EPC and other immune cells in-depth by single-cell RNA-sequencing of healthy and atherosclerotic mice aortas in our future study.

2. Authors use only LC3 as the autophagy markers of choice. I am not sure why they only use LC3 as the autophagy marker. Have you checked the other autophagy marker such as p62? Since LC3 goes up when pitavastatin treatment, the question arises whether all autophagy markers are going up or “flux”. If most markers are going up, then this is less consistent with

flux but rather it is a transcriptional response by TFEB, TFE3, or other master regulators. They can ascertain this by doing qPCR of the same markers to see if they go up. If the transcripts go up, then the authors are really not assessing “flux”. For “flux”, the authors will need to use agents such as Bafilomycin or chloroquine to see if LC3 and p62 change accordingly.

Responses: We are very grateful to your professional suggestions. Besides LC3 (MAP1LC3B), we as well used p62 (SQSTM1) to evaluate mitophagy in Figure 2A, Figure 3C, 3E and Figure 4A. Actually, as is shown in fig 3 G and H, keima-red was employed to verify the mitophagy flux and further exclude the possibility of reduced autophagosome turnover. According to your suggestion, we added the LC3 and p62 data with bafilomycin (BAFA1) and PTV treatment. The corresponding results were shown in revised manuscript (**Supplementary fig. 4**).

3. However, the big issue that arises here is again, why is pitavastatin, which affect mitophagy, having any effect on global autophagy? This piece of the paper needs to be explained and unraveled.

Responses: We are very grateful to your professional suggestions. Macroautophagy and mitophagy share partial molecular mechanisms and pathways, such as both the formation of autophagosomes and mitophagosomes needs MAP1LC3 cleavage and ubiquitination in phagophore and envelops the substrate to lysosomes. In this study, we employed MAP1LC3B(LC3) and SQSTM1(p62) to evaluate mitophagy, in addition, we found that mitophagy-related protein PINK1 and PARK2 decreased in EPC from atherosclerosis mice and mitophagy flux (measured by mt-keima-red vector) data confirmed mitophagy inhibition, which suggested that mitophagy was the major mechanism. PTV treatment increased MAP1LC3B-II, PINK1, PARK2 expression, but decreased p62 accumulation. Importantly, the vast majority of MAP1LC3B colocalized with mitochondria, and MAP1LC3B colocalized with PARK2. It is now generally believed that PINK1 was canonical mitophagy protein, and PINK1, PARK2, and ubiquitin have pivotal roles in priming mitophagy. However, the entire regulatory landscape and the precise control mechanisms of mitophagy remain to be elucidated. We used PINK1 knockout mice and PINK1 silencing method to disturb mitophagy to confirm that the benefit effect of PTV on EPC contributed to mitophagy reversion. Therefore, we considered that PTV protect EPC

dependent on mitophagy, rather than macroautophagy. According to your suggestion, we have discussed this issue in the section of Discussion in revised manuscript.

4. How is atherosclerosis reduced when pivalastatin is given? Any changes to atherosclerosis in pivalastatin administration is likely secondary effects unless the authors can show pivalastatin actually tangibly affects tissue levels.

Responses: Thank you. Indeed, pivalastatin could effectively reduce the level of serum cholesterol and was commonly used in clinical treatment. Besides, in this study, we have revealed another benefit effect on EPC, which contributed to repair of damage arterial endothelium. Arterial endothelium impairment is the initial process of atherosclerosis development. PTV protects EPC and promote repair of endothelium, which blocks atherosclerosis development. In tissue level, we accordingly showed that the PTV-pretreated EPC homed and localized on impaired arterial intima in laser scanning confocal microscopy (**Figure 10A**). Furthermore, Evans blue staining showed that PTV-pretreatment EPC increased reendothelialization of arterial intima, which was inhibited by sh*CAMK1* or sh*ATG7* (**Figure 10B**). The results above suggested that the actually protective effect of PTV on arterial endothelium during atherosclerosis.

Thank you and all the reviewers again for the positive comments and valuable suggestions.

Sincerely yours

References

- [1] Katayama H, Kogure T, Mizushima N, Yoshimori T, Miyawaki A. A sensitive and quantitative technique for detecting autophagic events based on lysosomal delivery. *Chem Biol*. 2011 Aug 26;18(8):1042-52.
- [2] Arao K, Yasu T, Umemoto T, Jinbo S, Ikeda N, Ueda S, Kawakami M, Momomura S. Effects of pitavastatin on fasting and postprandial endothelial function and blood rheology in patients with stable coronary artery disease. *Circ J*. 2009 Aug;73(8):1523-30.
- [3] Zu Y, Wan LJ, Cui SY, Gong YP, Li CL. The mitochondrial Na⁽⁺⁾/Ca⁽²⁺⁾ exchanger may reduce high glucose-induced oxidative stress and nucleotide-binding oligomerization domain receptor 3 inflammasome activation in endothelial cells. *J Geriatr Cardiol*. 2015 May;12(3):270-8.
- [4] Sirvent P, Mercier J, Vassort G, Lacampagne A. Simvastatin triggers mitochondria-induced Ca²⁺ signaling alteration in skeletal muscle. *Biochem Biophys Res Commun*. 2005 Apr 15;329(3):1067-75.
- [5] De Stefani D, Rizzuto R, Pozzan T. Enjoy the Trip: Calcium in Mitochondria Back and Forth. *Annu Rev Biochem*. 2016 Jun 2;85:161-92.
- [6] De Marchi E, Bonora M, Giorgi C, Pinton P. The mitochondrial permeability transition pore is a dispensable element for mitochondrial calcium efflux. *Cell Calcium*. 2014 Jul;56(1):1-13.

Reviewers' comments:

Reviewer #1 (Remarks to the Author):

In this revised manuscript, the authors clarified aspects of their approach and added important control data. While it may well require Ca²⁺ and Camk1 signaling to drive PINK1 stabilisation, the resulting autophosphorylation and downstream activation of Park2, a direct phosphorylation of Pink1 Ser228 or Park2 Ser65 by Camk1 must be further substantiated by clear-cut data. The in vitro kinase assay presented now does not provide compelling enough evidence that Pink1 (or Park2) is directly phosphorylated by Camk1.

As suggested before, authors should tone down their statements and focus on what the data truly shows while clearly highlighting caveats and limitations of their findings. Some of the current claims are still not backed up by the data presented. Instead they could simply acknowledge and state that the phosphorylation could be indirect as well.

With respect to supplementary figure 5 and the in vitro kinase assay.

Panels A and B seem to be flipped between figure and figure legend which makes it a bit more difficult to navigate. There also seems to be a disconnect between labels and actual molecular weight of the recombinant proteins used. Further the GAPDH blot is quite puzzling in the context of a pure in vitro assay using recombinant proteins only. Regardless of these flaws, the data does still not unequivocally support that Camk1 directly phosphorylates PINK1 at Ser228. What it perhaps does show though is that both Camk1 and a denatured PINK1 fragment can autophosphorylate themselves. While it seems surprising to see autophosphorylation of a denatured fragment, PINK1 autophosphorylation at Ser228 is well documented in the literature and seems to be important for its enzymatic function. This is neither acknowledged nor appropriately discussed especially in light of the authors' suggestion that Camk1 phosphorylates Ser228 of Pink1.

The Pink1 KO samples were certainly helpful and reassuring of the selectivity of the used Pink1 antibody, but the specificities of the phospho-antibodies to detect only the phosphorylated forms of Pink1 or Park2, but not the unphosphorylated proteins, remain to be established. While mitophagy was evaluated and seemed to be reduced in these Pink1 KO samples, in line with the literature, this was not necessarily analysed here in the context of their model and treatment.

Reviewer #3 (Remarks to the Author):

The authors have nicely improved the manuscript by adding figures based on comments, etc. The manuscript flows well now and is logical/focused. No further comments.

Reviewer #4 (Remarks to the Author):

The manuscript seems to be improved.

Minor issues still exist.

For example:

While the authors mentioned potential role of mitochondrial DNA mutations in Discussion section it would be nice and beneficial for the paper to have at least a brief mention about atherosclerosis, hypothesis or theories of the development of this disease including potential role of mitochondrial mutations in Introduction section.

Page 25 "In conclusion, as shown in the schematic drawing (Fig. 10), EPC proliferation and mitophagy are inhibited in atherosclerotic mice."

Figure 10 does not correspond to this sentence. The correct reference to the figure number should be provided.

The authors should check again their manuscript for the correctness of numbers of figures used.

"The role of statins as vasculoprotective agents are mostly attributed to their circulating cholesterol - lowering effect."

I think the sentence from the above still contains an error. It should be:

"The role of statins as vasculoprotective agents is mostly attributed to their circulating cholesterol - lowering effect."

The authors should check the manuscript for grammatical errors.

The manuscript can be accepted after minor revision.

Dear reviewer,

We would like to thank you for your constructive comments concerning our article entitled “Pitavastatin activates mitophagy to protect EPC proliferation through a calcium-dependent CAMK1-PINK1 pathway in atherosclerotic mice”. These comments are all valuable and helpful for improving our article. All the authors have seriously discussed about all these comments and tried best to modify our manuscript. Changes to our manuscript were all highlighted by using red colored text. Point-by-point responses to the reviewers are listed below this letter.

Reviewer #1 (Remarks to the Author):

In this revised manuscript, the authors clarified aspects of their approach and added important control data. While it may well require Ca²⁺ and Camk1 signaling to drive PINK1 stabilisation, the resulting autophosphorylation and downstream activation of Park2, a direct phosphorylation of Pink1 Ser228 or Park2 Ser65 by Camk1 must be further substantiated by clear-cut data. The *in vitro* kinase assay presented now does not provide compelling enough evidence that Pink1 (or Park2) is directly phosphorylated by Camk1.

1. As suggested before, authors should tone down their statements and focus on what the data truly shows while clearly highlighting caveats and limitations of their findings. Some of the current claims are still not backed up by the data presented. Instead, they could simply acknowledge and state that the phosphorylation could be indirect as well.

Responses: We are very grateful to your comments. We have modified the related statements that are not supported by existing data. The revised sentences are marked by red color in the manuscript resubmitted. Furthermore, we also included this point into the Discussion section.

2. With respect to supplementary figure 5 and the *in vitro* kinase assay.

Panels A and B seem to be flipped between figure and figure legend which makes it a bit more difficult to navigate. There also seems to be a disconnect between labels and actual molecular weight of the recombinant proteins used. Further the GAPDH blot is quite puzzling in the context of a pure *in vitro* assay using recombinant proteins only. Regardless of these flaws, the data does still not unequivocally support that Camk1 directly phosphorylates PINK1 at

Ser228. What it perhaps does show though is that both Camk1 and a denatured PINK1 fragment can autophosphorylate themselves. While it seems surprising to see autophosphorylation of a denatured fragment, PINK1 autophosphorylation at Ser228 is well documented in the literature and seems to be important for its enzymatic function. This is neither acknowledged nor appropriately discussed especially in light of the authors' suggestion that Camk1 phosphorylates Ser228 of Pink1.

Responses: Thanks for your kind suggestion. We feel really sorry for our carelessness in **supplementary figure 5**, and we have adjusted the related panels to the correct position. In the revised figures (now **supplementary figure 6**), we have corrected the mislabeled molecular weight of recombinant proteins, and deleted the misplaced GAPDH blot. We have acknowledged that we cannot provide enough evidence to confirm that CAMK1 directly phosphorylates PINK1 at Ser²²⁸ in revised manuscript. We consider that CAMK1 activation is associated with PINK1 phosphorylation, And, we added the effect of PINK1 autophosphorylation on its function in Discussion section.

3. The Pink1 KO samples were certainly helpful and reassuring of the selectivity of the used Pink1 antibody, but the specificities of the phospho-antibodies to detect only the phosphorylated forms of Pink1 or Park2, but not the unphosphorylated proteins, remain to be established.

Response: Thanks for your comment. As shown in **Figure 6A**, we indeed detected a clear single band with higher molecular weight when using the phospho-antibodies. And, it only account for a small part of the total PINK1 or PARK2 proteins (compared to the right bolts), which is consistent with the results in other researches using the same phospho-site antibody¹⁻². Therefore, we believed that the detected band could be regarded as the phosphorylated form of PINK1 or PARK2.

4. While mitophagy was evaluated and seemed to be reduced in these Pink1 KO samples, in line with the literature, this was not necessarily analysed here in the context of their model and treatment.

Responses: We replaced this part to the supplementary material (**supplementary figure 5**).

Reviewer #4 (Remarks to the Author):

1. While the authors mentioned potential role of mitochondrial DNA mutations in Discussion section it would be nice and beneficial for the paper to have at least a brief mention about atherosclerosis, hypothesis or theories of the development of this disease including potential role of mitochondrial mutations in Introduction section.

Responses: Thanks for your suggestion. We have added this point in the Introduction section in the revised manuscript.

2. Page 25 “In conclusion, as shown in the schematic drawing (Fig. 10), EPC proliferation and mitophagy are inhibited in atherosclerotic mice.” Figure 10 does not correspond to this sentence. The correct reference to the figure number should be provided.

Responses: We have corrected it into **Figure 11**, and rechecked all figure numbers carefully.

3. The authors should check again their manuscript for the correctness of numbers of figures used.

“The role of statins as vasculoprotective agents are mostly attributed to their circulating cholesterol -lowering effect.”

I think the sentence from the above still contains an error. It should be:

“The role of statins as vasculoprotective agents is mostly attributed to their circulating cholesterol -lowering effect.”

The authors should check the manuscript for grammatical errors.

Responses: Thanks for your careful correction. And we have carefully screened the manuscript and rechecked the grammatical errors in the full text. The revised sentences are marked by red color in the manuscript resubmitted.

Thank you and all the reviewers again for the positive comments and valuable suggestions.

Sincerely yours

References

1. Kato Y, Sakamoto K. Niclosamide affects intracellular TDP-43 distribution in motor neurons, activates mitophagy, and attenuates morphological changes under stress [published online ahead of print, 2021 Aug 21]. *J Biosci Bioeng.* 2021;S1389-1723(21)00178-X.
2. Miyai T, Vasanth S, Melangath G, et al. Activation of PINK1-Parkin-Mediated Mitophagy Degrades Mitochondrial Quality Control Proteins in Fuchs Endothelial Corneal Dystrophy. *Am J Pathol.* 2019;189(10):2061-2076.

REVIEWERS' COMMENTS:

Reviewer #1 (Remarks to the Author):

The authors finally acknowledge some of the limitations of their experiments and point out the uncertainty of a direct phosphorylation of Pink1 by Camk1.

Reviewer #4 (Remarks to the Author):

The manuscript was improved.

But still, there are some issues left.

In methods section it is not clearly stated where do apoE^{-/-} mice originate from. It should be corrected.

In introductory section the authors still did not explain clearly what is atherosclerosis. They added sentences devoted to the role of mitochondrial mutations in atherosclerosis development but the reference #16 used in this section does not contain any mention of the role of mitochondrial mutations in ECs and VSMCs.

"In the initial stages of atherosclerosis, mutant and/or damaged mitochondria accumulated in endothelial cells (ECs) and vascular smooth muscle cells (VSMCs)¹⁶, which resulted in EC dysfunction and altered VSMCs phenotype. "

Also, they did not mention about another potential inductors of atherosclerosis development, desialylated lipoproteins and oxidized lipoproteins (for example, the role of desialylated lipoproteins in atherosclerosis was recently reviewed in DOI: 10.3390/biomedicines9060600). Otherwise it will look like mitochondrial mutations may be the only one potential reason for atherosclerosis induction.

Some sentences still need a correction of grammar:

"In the initial stages of atherosclerosis, mutant and/or damaged mitochondria accumulated in endothelial cells (ECs) and vascular smooth muscle cells (VSMCs)¹⁶, which resulted in EC dysfunction and altered VSMCs phenotype. "

"In the end, this driven vulnerable plaque phenotype and promote atherosclerosis through inflammation, oxidative stress, and altering lipid metabolism process¹⁷. "

"The defective clearance of dysfunctional mitochondria caused by mtDNA mutations further results in an accumulation of dysfunctional mitochondria, ROS overproduction, cellular disorder and even apoptosis, and promoting atherosclerosis⁴²."

The authors need to correct their manuscript for grammatical errors.

Minor revision is required.

Dear reviewer,

We appreciated your constructive comments concerning our article entitled “Pitavastatin activates mitophagy to protect EPC proliferation through a calcium-dependent CAMK1-PINK1 pathway in atherosclerotic mice”. The comments are helpful for improving our article. We have revised our manuscript according to your suggestion. New changes to our manuscript were highlighted using red colored text. Point-by-point responses to the reviewers are listed below.

Reviewer #4 (Remarks to the Author):

1. In methods section it is not clearly stated where do apoE^{-/-} mice originate from. It should be corrected.

Responses: Thanks for your reminding. We have added the source of apoE^{-/-} mice in the methods section.

2. In introductory section the authors still did not explain clearly what is atherosclerosis. They added sentences devoted to the role of mitochondrial mutations in atherosclerosis development but the reference #16 used in this section does not contain any mention of the role of mitochondrial mutations in ECs and VSMCs.

“In the initial stages of atherosclerosis, mutant and/or damaged mitochondria accumulated in endothelial cells (ECs) and vascular smooth muscle cells (VSMCs)¹⁶, which resulted in EC dysfunction and altered VSMCs phenotype.”

Responses: Thanks for your kind suggestion. We feel sorry for our carelessness in **reference #16**, and we have replaced it by the correct reference (now **reference #17**). Additionally, we added the statement about atherosclerosis in Introduction section in the manuscript resubmitted (red words).

3. Also, they did not mention about another potential inductors of atherosclerosis development, desialylated lipoproteins and oxidized lipoproteins (for example, the role of desialylated lipoproteins in atherosclerosis was recently reviewed in DOI: 10.3390/biomedicines9060600). Otherwise it will look like mitochondrial mutations may be the only one potential reason for atherosclerosis induction.

Responses: We added other inductors besides mitochondrial mutations during atherosclerosis development and cited related research in the revised manuscript (now **reference #16**).

4. Some sentences still need a correction of grammar:

Responses: Thank you. And we have carefully rechecked the grammatical errors in the corresponding paragraph. The revised sentences are marked by red color in the revised manuscript

“In the initial stages of atherosclerosis, mutant and/or damaged mitochondria accumulated in endothelial cells (ECs) and vascular smooth muscle cells (VSMCs)¹⁶, which resulted in EC dysfunction and altered VSMCs phenotype. ”

Revised: In the early stage of atherosclerosis, increased production of reactive oxygen species (ROS) in mitochondria, accumulation of mitochondrial DNA damage, and progressive respiratory chain dysfunction, resulted in endothelial cells (ECs) dysfunction and vascular smooth muscle cells (VSMCs) phenotypic conversion¹⁷

“In the end, this driven vulnerable plaque phenotype and promote atherosclerosis through inflammation, oxidative stress, and altering lipid metabolism process¹⁷. ”

Revised: After a long term, ECs apoptosis, VSMCs phenotypic conversion and inflammatory cells infiltration further promoted the development of atherosclerosis and led to vulnerable plaque in the end.

“The defective clearance of dysfunctional mitochondria caused by mtDNA mutations further results in an accumulation of dysfunctional mitochondria, ROS overproduction, cellular disorder and even apoptosis, and promoting atherosclerosis⁴².”

Revised: MtDNA damage and deletion lead to reduced complex I activity and decreased oxidative phosphorylation, which further increased mitochondrial dysfunction. The accumulation of damaged mitochondria further results in ROS overproduction, cellular disorder and even apoptosis⁴⁰

Thank you and all the reviewers again for the positive comments and valuable suggestions.

Sincerely yours